



# The boundary condition for the vertical velocity and its
# interdependence with surface gas exchange
Andrew S. Kowalski[1,2]
[1]Departmento de Física Aplicada, Universidad de Granada, Granada, 18071, Spain
[2]Instituto Interuniversitario de Investigación del Sistema Tierra en Andalucía, Centro Andaluz de Medio
Ambiente (IISTA -CEAMA), Granada, 18071, Spain
*Correspondence to*: Andrew S. Kowalski (andyk@ugr.es)
**Abstract.** The law of conservation of linear momentum is applied to surface gas exchanges, employing
scale analysis to diagnose the vertical velocity ($w$) in the boundary layer. Net upward momentum in the
surface layer is forced by evaporation ($E$) and defines non-zero vertical motion, with a magnitude defined
by the ratio of $E$ to the air density, as $w = \frac{E}{\rho}$. This is true even right down at the surface where the
boundary condition is $w|_0 = \frac{E}{\rho}|_0$. This Stefan flow velocity implies upward transport of a non-diffusive
nature that is a general feature of the troposphere but is of particular importance at the surface, where it
assists molecular diffusion with upward gas migration (of $H_2O$, e.g.) but opposes that of downward-
diffusing species like $CO_2$ during daytime. The definition of flux-gradient relationships (eddy
diffusivities) requires rectification to exclude non-diffusive transport, which does not depend on scalar
gradients. At the microscopic scale, the role of non-diffusive transport in the process of evaporation from
inside a narrow tube – with vapour transport into an overlying, horizontal air stream – was described long
ago in classical mechanics, and is routinely accounted for by chemical engineers, but has been neglected
by scientists studying stomatal conductance. Correctly accounting for non-diffusive transport through
stomata, which can appreciably reduce net $CO_2$ transport and marginally boost that of water vapour,
should improve characterizations of ecosystem and plant functioning.

## 1 Introduction
The vertical velocity ($w$) is a key variable in the atmospheric sciences, whose precise diagnosis is
essential for numerous applications in meteorology. Above the boundary layer, the weather is largely
determined by adiabatic adjustments to vertical motion that is slight compared to horizontal winds. Closer
to the surface, even a tiny $w$ can result in relevant transport; for example, in a typical boundary layer –
with representative temperature ($T = 298K$), pressure ($p = 101325$ Pa), and $CO_2$ mass fraction (607 mg
$kg^{-1}$; a molar ratio of about 400 ppm) – just 61 µm $s^{-1}$ of average vertical velocity is needed to waft a
biologically significant 44 µg$CO_2$ $m^{-2}$ $s^{-1}$ (a $CO_2$ molar flux density of 1 µmol $m^{-2}$ $s^{-1}$). Modern
anemometry cannot resolve such miniscule airflow (Lee, 1998), and generally $w$ is immensurable at many
scales so that it must be derived from other variables (Holton, 1992). Such diagnostic estimation is
traditional in synoptic meteorology, but has been developed less rigorously near the surface boundary.





The characterization of boundary conditions for state and flow variables, in order to enable atmospheric
modelling at larger scales, is a fundamental goal of micrometeorology. Since $w$ is an air velocity, its
boundary condition $w|_0$ describes the surface-normal or vertical motion of the gas molecules found
closest to the surface (at some height $z|_0$, very nearly but not exactly zero). The Navier-Stokes equations,
when applied to the lower atmosphere, are particularly sensitive to the conditions specified at the
boundary (Katul et al., 2004), and this lends great importance to $w|_0$ in the context of dynamic modelling.
Nevertheless, until now $w|_0$ has received inadequate attention in boundary-layer meteorology.

Micrometeorologists have made presuppositions regarding $w|_0$ without formal justification and in
contradiction to deductions from classical mechanics. The traditional hypothesis about near-surface winds
is that they flow parallel to underlying terrain (Kaimal and Finnigan, 1994;Wilczak et al., 2001) and
vanish at the surface (Arya, 1988), implying $w|_0 = 0$. This assumption underlies many derivations and
abets the prevailing belief that vertical exchanges are accomplished purely by molecular diffusion within
a millimeter of the surface (Foken, 2008), or purely by turbulent diffusion at heights of meters or more
within the atmospheric boundary layer. However, such a premise is inconsistent with the fact of net
surface gas exchange (predominantly evaporative), which implies Stefan flow with a mean velocity
component normal to the surface.

Net mass transfer across a surface results in a velocity component normal to the surface, and an
associated non-diffusive flux in the direction of mass transfer (Kreith et al., 1999). The existence and
relevance of Stefan flow – first derived and described in the $19^{th}$ century – is certain. Indeed, engineers
necessarily account for its role in heat and mass transfer (Abramzon and Sirignano, 1989) when precisely
controlling industrial processes that include phase change, such as combustion. For these reasons, it is to
be expected that a more accurate means of estimating $w|_0$ for the atmospheric boundary layer can be
achieved by rigorous examination of known surface flux densities in the light of physical laws.

The remaining sections of this work aim to diagnose a defensible lower boundary condition for the
vertical velocity $(w|_0)$ and to interpret its significance. Section 2 presents the theory, and illustrates types
of mass transport and heat exchange in fluids via an example from the liquid phase. In Section 3, an
analytical framework is established and conservation of linear momentum is applied to derive $w|_0$ from
published magnitudes of surface gas exchanges, demonstrating that it is directly proportional to the
evaporative flux density ($E$), consistent with the findings of Stefan. The derived vertical velocity is seen
to be relevant in defining the mechanisms of gas transport, which is not accomplished by diffusion alone
– even at the surface interface. Section 4 highlights the need to rectify flux-gradient relationships by
taking into account the non-diffusive component of transport; this includes boundary-layer similarity
theory and physiological descriptions of stomatal conductance. Thus, the implications of these analyses
are broad and interdisciplinary.





## 2 Theory

The objective of this section is to establish the theoretical bases for the analyses and interpretations that follow. It opens with a list of symbols (Table 1) along with the meaning and S.I. units of each variable represented, and finishes with a summary of the most salient points regarding physical laws and transport mechanisms to be recalled in Section 3.

### 2.1 Relevant Scientific Laws

#### 2.1.1 The Law of Conservation of Linear Momentum

The principle of conservation of momentum is most fundamental in physics, more so than even Newton's 1st Law (Giancoli, 1984). It defines the momentum of a system of particles as the sum of the momenta of the individual components, and establishes that this quantity is conserved in the absence of a net external force. Accordingly, in atmospheric dynamics (Finnigan, 2009) a system may be defined as the $N$ component gas species comprising a particular mass of air, with a net vertical momentum flux density of

$$w\rho = \sum_{i=1}^{N} w_i \rho_i. \tag{1}$$

In Eq. (1), $w$ and $\rho$ represent the velocity and density of air, respectively, while $w_i$ and $\rho_i$ are the those properties of component $i$, whose species flux density is $w_i \rho_i$. For this species $i$, total transport $w_i \rho_i$ can be attributed to mechanisms that are diffusive (if $w_i \neq w$), non-diffusive (if $w \neq 0$), or more generally a combination of these two types of transport. Dividing Eq. (1) by the net air density defines the system's vertical velocity as a weighted average of those of its components (Kowalski, 2012), where the weighting factors are the species' densities.

#### 2.1.2 The 0th Law of Thermodynamics

The 0th Law establishes the temperature as the variable whose differences determine the possibility for heat exchange between thermodynamic systems. For two systems in thermal contact, if they have the same temperature then they are in thermodynamic equilibrium and therefore exchange no heat. If their temperatures differ, then heat will be transferred from the system with the higher temperature to that with the lower temperature. Heat transfer by molecular conduction depends on gradients in the temperature; in compressible fluids like air, however, turbulent diffusion can occur without thermal contact and yet bring about heat transfer as determined by gradients in the potential temperature (Kowalski and Argüeso, 2011), accounting for any work done/received during the expansion/compression associated with vertical motions.

#### 2.1.3 Fick's 1st Law of Diffusion

Molecular diffusion has no effect on the net fluid momentum, but "randomly" redistributes fluid components and can cause different species to migrate in different directions, according to component scalar gradients. Regrettably, scientific literature contains inconsistencies regarding the scalar whose gradient determines diffusion in the gas phase (Kowalski and Argüeso, 2011). The proper form of Fick's 1st Law for diffusion in the vertical direction is





$F_{i,M} = -\rho k \frac{\partial f_i}{\partial z}$,                                                                                    (2)
where $F_{i,M}$ is the vertical flux density of species $i$ due to molecular diffusion, which is proportional to the
vertical gradient in that species' mass fraction ($f_i$; Bird et al. (2002)), and $z$ is height. Also relevant are the
fluid density ($\rho$) and molecular diffusivity ($k$).  However, $\rho$ must not be included in the derivative in Eq.
(2), unless for the trivial case where it is constant (as in an incompressible fluid); in compressible media,
gradients in gas density can arise, with no direct relevance to diffusion, due to gradients in pressure or
temperature as described by the Ideal Gas Law. It is relevant to note that Adolf Fick arrived at this law,
not by experimentation, but rather by analogy with Fourier's law for heat conduction (Bird et al., 2002).
By the same analogy, the product of the diffusivity with the scalar gradient in Eq. (2) yields a kinematic
flux, which requires multiplication by the fluid density in order to yield the flux density of interest.

Fluxes due to molecular diffusion are referenced to the motion of the fluid's centre of mass, or "mixture
velocity" (Bird et al., 2002). The simplest example to describe this is that of binary diffusion where only
two species compose the fluid, as in the traditional meteorological breakdown of air into components
known as dry air and water vapour. In the case of "static diffusion", the fluid velocity is zero and the mass
flux of one gas species (water vapour) counterbalances that of the other (dry air). When diffusion occurs
in a dynamic fluid (non-zero velocity), then overall transport must be characterized as the sum of
diffusive and non-diffusive components.

Turbulent diffusion is analogous to molecular diffusion in the sense that fluid components are randomly
redistributed, with different species migrating as a function of gradients in their mass fractions. The
primary difference is that eddies rather than molecular motions are responsible for mixing, and the eddy
diffusivity (the value of $k$ in Eq. (2), describing "K-theory" (Stull, 1988)) is a property of the flow rather
than the fluid. The Reynolds number describes the relative importance of molecular and turbulent
diffusion, which are otherwise indistinct with respect to the analyses that follow, and will simply be
grouped and referred to as "diffusive transport".

**2.2 Transport processes**
In this section, two case studies from the liquid phase will help identify and define non-diffusive and
diffusive types of transport, as well as their scalar source/sink determinants. Let us consider the case of
freshwater ($35 \cdot 10^{-5}$ mass fraction of salt) with constant temperature and composition flowing through a
tube into the bottom of a pool (Figure 1). Considering only flow within the tube (at point 1), whether
laminar or turbulent, it clearly realizes non-diffusive transport of salt, since the salt has no particular
behaviour with respect to the fluid, but simply goes with the flow. There are no scalar gradients within the
tube, and so there is neither diffusion nor advection. Let us now describe diffusive transport processes
within the pool (at point 2), and the nature (whether absolute or relative) of the relevant fluid properties
whose gradients determine them by defining sources/sinks, using two illustrative case scenarios.





The temperature is constant in time and space, but other *characteristics of the two case scenarios* are
chosen to elucidate the relationship between diffusive transport processes and scalar gradients:
1)  *Due to surface evaporation that balances the mass input from the tube, the pool mass is constant; the*
*water is maintained isothermal by surface heating that supplies the (latent) energy for evaporation.*
*Initially ($t_0$) the pool has zero salt mass, but salinity increases constantly, equalling that of the tube*
*water at some moment ($t_{eq}$) and rising by another two orders of magnitude to reach that of sea water*
*($35 \cdot 10^{-3}$) by the end of the scenario ($t_f$).* This case is of interest from both salt/solute and
thermodynamic points of view:
a)  In solute terms, the tube represents a source of (absolute) salt to the pool, but not always of
(relative) salinity. Initially *($t_0$)*, the water from the tube is more saline than that in the pool, such
that non-diffusive and diffusive transport processes operate in tandem to transport salt from the
tube upward into the pool; at this moment, the tube is a source of salinity. Salinity advection,
defined as the negative of the inner product of two vectors (the velocity with the salinity
gradient, with opposite signs), is then positive. Ultimately however (at $t > t_{eq}$), the water in the
pool is more saline than that entering from the tube, such that non-diffusive and diffusive salt
transport are in opposite directions; then the tube dilutes the pool and is a salinity sink, but still a
salt source. Salinity advection at $t_f$ is negative. The pool continues to gain salinity after $t_{eq}$,
despite the diluting effects of the tube, due to the concentrating effects of evaporation, which is
the ultimate source of salinity. This distinction matters because the gradients that drive advection
and diffusion are those in salinity, a relative (not absolute) salt measure. At $t_f$, the diffusive
salinity fluxes are oriented against the flow within the pool (downward, and radially inward
towards the diluting tube, despite its being a net salt source). By contrast, non-diffusive transport
always goes with the flow, and accounts for continued upward and outward salt transport,
increasing the salt content at the surface.
b)  Although thermodynamically trivial – with no heat exchanges whatsoever within the water as
determined by the 0th Law – this case nonetheless illustrates the nature of the scalars that
determine heat transfer by advection and diffusion (conduction). The "heat content" of the pool
decreases as it becomes more and more saline, due to the inferior heat capacity of saltwater
versus freshwater. Similarly, salt diffusion/advection is initially upward/positive but ultimately
downward/negative, yet the corresponding implications regarding heat content fluxes say
nothing about the transfer of heat. The point here is that the dynamics of the heat content must
not be interpreted in terms of heat fluxes, which was done by Finnigan et al. (2003). For this
reason, meteorologists correctly define "temperature advection" (Holton, 1992) based the
thermodynamic relevance of gradients in the variable singled out by the 0th Law.

2)  *Let us now specify that the water in the pool has the same (freshwater) salinity as that coming from*
*the tube ($35 \cdot 10^{-5}$). If we furthermore remove both surface evaporation and heating from scenario (1),*
*then the temperature remains constant and the salinity corresponds uniformly to that of freshwater,*
*but the pool accumulates mass.* In this case, there are convergences in the non-diffusive transports of
water, salt, and heat content: fluxes into the pool are positive, while fluxes out are null. However,



there are no gradients in temperature or salinity, and so there is neither diffusion nor advection in this
scenario. The pool does gain volume (depth) but this is only because the fluid under consideration is
incompressible. By contrast, for the gas phase, accumulation of absolute quantities – such as air and
trace constituent mass, and heat content – can occur in a constant volume context (e.g., "at a point")
due to convergent, non-diffusive transport that defines compression. In the pool, diffusion and
advection are clearly null because they are determined by gradients in the relative trace gas amount –
the mass fraction –, a variable of essential utility for the gas phase because it is immune to the effects
of compression.

**2.3 An advection-diffusion synopsis**


The analyses that follow rely on the succeeding key points drawn from sections 2.1 and 2.2. Advection
and diffusion depend on gradients in scalars whose nature is relative rather than absolute. In
incompressible thermodynamics, the relevant gradients are those in the temperature, and not the heat
content. For trace constituents, the relevant scalar is the mass fraction (e.g., salinity) and not the species
density. Advection and diffusion are otherwise physically very distinct. Like non-diffusive transport,
diffusion is a vector whose vertical component is of particular interest in the context of surface-
atmosphere exchange. By contrast, advection is a scalar; for some arbitrary quantity $\xi$, it is defined as the
negative of the inner product $\mathbf{v} \cdot \nabla \xi$, where $\mathbf{v}$ is the fluid velocity and $\nabla$ is the gradient operator. To be
clear, it can make sense to speak of "upward diffusion", but certainly not "upward advection". The
tendency, in the science of surface-atmosphere exchange, to speak of "vertical advection" (e.g., Rannik et
al., 2009) is intimately related to an assumption of horizontal homogeneity, precluding horizontal scalar
gradients particularly in the direction of the mean wind.

The scenarios depicted above correspond to the incompressible case (liquid). When the effects of
compressibility are irrelevant, it can be convenient to add the incompressible form of the continuity
equation ($\nabla \cdot \mathbf{v} = 0$) to advection yielding $-\nabla \cdot \xi \mathbf{v}$, the convergence of a kinematic flux. This is called the
"flux form" of advection. For a compressible medium such as the atmosphere, however, if $\xi$ is taken to
represent some "absolute fluid property such as the (gas) density" (Finnigan et al., 2003), then the
transformation of advection into flux form cannot be justified (Kowalski and Argüeso, 2011), since using
the incompressible form of the continuity equation leads to unacceptable errors in conservation equations
for boundary-layer control volumes (Kowalski and Serrano-Ortiz, 2007). By contrast, the expression of
advection in flux form can be valid if the scalar $\xi$ is carefully chosen for its immunity to the effects of
compression, as is the case for the mass fraction. These generalizations regarding the nature of transport
by non-diffusive and diffusive mechanisms, and also the nature of advection, will now be applied to the
case of vertical transport very near the surface and the mechanisms that participate in surface exchange,
after first deriving the boundary condition $w|_0$.





**3 Analysis**
**3.1 Framework**
The analysis will focus on a system defined as a mixture of gas molecules of different species, whose
momentum will be examined. The system's mass is defined (Table 2) by gas components in a ratio that
corresponds quite closely to that of the atmosphere (Wallace and Hobbs, 2006) but updated to more
closely reflect actual atmospheric composition. At a representative ambient temperature ($T$ = 298 K) and
pressure ($p$ = 101325 Pa), the many millions of molecules forming this system occupy a volume of $10^{-15}$
m$^3$ with 70% relative humidity. The system geometry will be specified in four different ways, according
to the different spatial scales for which $w|_0$ is to be described:

231        A.  At the synoptic scale, the volume occupied by the system is a lamina of depth $\delta z \sim 10^{-27}$ m,

bounded above and below by constant geopotential surfaces, with horizontal dimensions ($\Delta x$ and

$\Delta y$) on the order of $10^6$ m. The fact that $\delta z$ is thinner than the dimension of a molecule matters

not at all when classifying any and all molecules whose centres of mass (points, with neither size

nor dimension) occupy the lamina as belonging to the volume;

B.  At the micrometeorological scale, the volume overlies a flat surface and is shaped as a

rectangular lamina of depth $\delta z \sim 10^{-21}$ m, with horizontal dimensions ($\Delta x$ and $\Delta y$) of $10^3$ m;

C.  At the leaf scale, the volume is a rectangular lamina of depth $\delta z \sim 10^{-11}$ m, with horizontal

dimensions ($\Delta x$ and $\Delta y$) of $10^{-2}$ m; and

D.  At the microscopic scale of plant stomata, the volume is a cube with $\Delta x = \Delta y = \delta z = 10^{-5}$ m. For

the purpose of transitioning between the leaf and microscopic scales, plant pores are assumed to

occupy a stomatal fraction $\sigma$ of the leaf surface and yet accomplish all gas exchange, with the

remaining fraction (1-$\sigma$) occupied by a cuticular surface whose gas exchange is assumed to be

null (Jones, 1983).

Independent of scale, the base height $z|_0$ of the volume is the lowest for which only air – and neither
ocean wave nor land surface element – occupies the volume. The land/ocean/leaf surface will be assumed
to be static (i.e., its vertical velocity is zero), impenetrable to the wind (explicitly neglecting ventilation of
air-filled pore space), smooth, level and uniform, all for the sake of simplicity. The temporal framework
for the analysis is instantaneous, with no need to choose between Eulerian and Lagrangian fluid
specifications.

The direction of momentum transport to be examined is vertical, meaning perpendicular to constant
geopotential surfaces and therefore to the underlying surface. At the stomatal scale, the stoma to be
examined is situated on the upper side of a flat, horizontal leaf; water vapour exiting the stomatal aperture
during transpiration therefore has a positive vertical velocity. These analyses can be generalized to
sloping surfaces and/or stomata on the underside of leaves, simply by referring to the "surface-normal"
rather than "vertical" velocity. Hereinafter, however, the term "vertical" will be employed for
conciseness.





**3.2 The vertical velocity at the surface boundary**
Knowledge regarding surface exchange (gas flux densities) has advanced to the point where the boundary
condition for the vertical velocity $(w|_0)$ can be estimated from conservation of linear momentum –
applying Eq. (1) to the system defined in Table 2 –, and vastly simplified to a simple function of the
evaporation rate ($E$).  The species flux densities $(w_i \rho_i)$  within the system represent the surface exchanges
of the corresponding gas species ($i$).  Scale analysis of surface gas exchange magnitudes, published from
investigations at a particularly well-equipped forest site in Finland (Table 3), reveals that for the water
vapour species ($i$=4), the flux density $(E = w_4 \rho_4)$ is orders of magnitude larger than both the flux density
of any dry air component species and even the net flux density of dry air. Such dominance by water
vapour exchanges is representative of most surfaces worldwide. This is especially so because the two
largest dry air component fluxes are opposed, with photosynthetic/respiratory $CO_2$ uptake/emission
largely offset by $O_2$ emission/uptake (Gu, 2013). Hence, following tradition in micrometeorology (Webb
et al., 1980), dry air exchange can be neglected, allowing the elimination from Eq. (1),  when applied at
the surface, of all species flux densities except for that of water vapour ($H_2O$; $i$=4). Therefore, net air
transfer across the surface can be approximated very accurately as
$$w|_0 \, \rho|_0 = w_4|_0 \, \rho_4|_0 = E \; ,$$   (3)
where $w_4|_0$ and $\rho_4|_0$ are the $H_2O$ species velocity and density at the surface.  Equation (3) states that, at
the surface, the net vertical momentum flux density of air is equal to the net vertical momentum flux
density of water vapour, which is the evaporation rate. Solving this for $w|_0$  allows estimation of the
lower boundary condition for the vertical velocity as
$$w|_0 = \frac{E}{\rho_4|_0} \; .$$   (4)

The representative evaporation rate prescribed in Table 3 is valid for most of the scales defined above. In
the context of scale analysis, leaves may be approximated as having equal area as the underlying surface
(i.e., a unit leaf area index, or LAI=1), and equal evaporation rates as the surface in general. This latter
assumption does not neglect soil evaporation, but only excludes the possibility that it dominate leaf
evaporation by an order of magnitude. Thus, it will be assumed here that the assumed evaporation rate
and derived vertical velocities are equally valid at synoptic (A), micrometeorological (B), and leaf (C)
scales. However, for the microscopic (D) scale, it will be assumed that all leaf evaporation (or
transpiration) occurs through the small fraction of the leaf that is stomatal ($\sigma$), such that both the stomatal
evaporative flux density and the lower boundary condition for the vertical velocity $(w|_0)$ are a factor $1/\sigma$
greater than that at larger scales. Independent of scale, Eq. (4) states that, for a positive evaporation rate,
the boundary condition for the vertical velocity is non-zero and upward.

Given that the surface boundary is static, it may well be asked why there is a non-zero boundary condition
for the vertical velocity of air. The answer is that evaporation induces a pressure gradient force that
pushes air away from the surface. Evaporation into air increments the water vapour pressure and thereby



the total pressure, according to Dalton's law. If evaporation were to proceed until achieving equilibrium,
the pressure added by evaporation would correspond to the saturation vapour pressure ($e_s$; Figure 2),
whose temperature dependency has been quantified empirically and is described by the Clausius-
Clapeyron relation. It is this evaporation-induced pressure gradient force that pushes the manometer in
Fig. 2 to its new position, and similarly that drives winds away from the surface.

Although this upward air propulsion occurs at the surface, air velocities are generally upward throughout
the boundary layer in a climatological context. Indeed, the dominant role of water vapour in determining
the net vertical momentum of air is a general feature of the troposphere. In the context of the
hydrological cycle, water vapour is transported from the surface where it has an evaporative source, to
further aloft where clouds develop via processes that act as water vapour sinks: condensation and vapour
deposition onto ice crystals (or ice nuclei). In terms of total water, upward transport in the gas phase is
offset, over the long term, by downward transport in liquid and solid phases (e.g., rain and snow); unlike
the water vapour flux, however, precipitation does not directly define air motion. It is true that downward
water vapour transport occurs during dewfall – with surface condensation, as described by Eq. (4) with a
negative evaporation rate ($E < 0$) –, but this plays a minor role in the global water balance. Generally, the
relative magnitudes of gas exchanges used for the scale analysis in Table 3 are representative throughout
most of the troposphere, with upward water vapour flux densities dominating those of other gases in the
vertical direction. In the surface layer, sometimes termed the "constant flux layer" (Dyer and Hicks,
1970), Eq. (4) can be extrapolated away from the surface under steady-state conditions to yield
$$w = \frac{E}{\rho}. \hspace{5cm} (5)$$

### 3.3 Mechanisms of gas transport at the surface

Non-zero vertical momentum in the lower atmosphere and right at the surface boundary – dominated by
the flux density of water vapour and generally upward due to evaporation – means that diffusion is not the
lone relevant transport mechanism that participates in surface exchange, as has been generally supposed.
This is true for all atmospheric constituents, and not only for water vapour; over an evaporating surface,
any molecule undergoing collisions with its neighbours does not experience a random walk (a
characteristic of static diffusion), but rather tends to be swept upward with the flow. The upward air
current similarly wafts aerosol particles, although these may move downwards if their fall velocities
exceed the upward air motion. The upward flow velocity is rather small – just 31 $\mu$m s$^{-1}$ for the conditions
specified above and the evaporation rate of Table 3, according to Eqs. (4) and (5). It does not exclude the
possibility of diffusive transport in any direction, but does imply a relevant, non-diffusive component of
transport for any gas, whose magnitude is not related to that gas's scalar gradient.

The non-diffusive flux density of species $i$ can be expressed as
$$F_{i,non} = w\rho_i , \hspace{5cm} (6)$$



and when substituting for $w$ from Eq. (5) this becomes
$F_{i,non} = E f_i$ ,                                                                                     (7)
i.e., the product of the evaporation rate and the species mass fraction. Examination of its magnitude near
the surface for different gases will now show that, while this is often small in comparison with the
diffusive component, it is not negligible in every case, depending on the magnitudes of the mass fraction
and surface exchange for the gas considered.

Interpreting decomposed transport is simplest when examining a gas whose surface exchange is very well
known, such as the null value for inert Argon (Ar) that constitutes ca. 1.3% of dry atmospheric mass
(Wallace and Hobbs, 2006). Considering the state variables defined by Table 2 and the evaporation rate
of Table 3, Eq. (7) indicates 458 μg m$^{-2}$ s$^{-1}$ (a molar flux density of 11.6 μmol m$^{-2}$ s$^{-1}$) of upward, non-
diffusive Ar transport ($F_{3,non}$). To comprehend this, it helps to recall that the constant addition of $H_2O$
dilutes dry air at the surface and promotes its downward diffusion. For a null net flux of inert Ar to exist,
downward diffusion of this dry air component must exactly cancel the upward non-diffusive transport,
and therefore is 458 μg m$^{-2}$ s$^{-1}$ for the state and evaporative conditions specified above. These opposing
non-diffusive and diffusive Ar transport processes are quite analogous to case scenario 1 of Section 2.2, at
the instant $t_f$ when the fluid emitted into the pool has a diluting effect. Such dual transport mechanisms
are also relevant for vital gases, with different transport directions and degrees of relevance, depending on
the density and flux density of the gas in question.

For $H_2O$, the two types of gas transport mechanisms operate in tandem, with the non-diffusive component
contributing a fraction of upward $H_2O$ transport that, according to Eq. (7), is exactly the water vapour
mass fraction or specific humidity (Wallace and Hobbs, 2006)
$q \equiv f_4 \equiv \frac{\rho_4}{\rho}$ .                                                               (8)
This is just 2% for the state conditions previously specified, but can approach 5% for very warm
evaporating surfaces and/or high-altitude environments. The breakdown of $H_2O$ transport into diffusive
and non-diffusive components is analogous to case scenario 1 of Section 2.2 at an instant prior to $t_{eq}$ when
the fluid introduced to the pool is highly concentrated, in comparison with the fluid already in the pool. In
any case, non-diffusive $H_2O$ transport is generally secondary to diffusive transport, but its neglect in an
ecophysiological context can lead to larger relative errors, as will be shown in Section 4.

For $CO_2$, which usually migrates downward during evaporative conditions because of photosynthetic
uptake, upward transport of a non-diffusive nature is even more relevant, opposing the downward flux
due to diffusion. To see this, let us examine the typical gas transport magnitudes of Table 2 and the
atmospheric state conditions specified above. According to Eq. (7), non-diffusive $CO_2$ transport ($F_{5,non}$) is
then 21.5 μg m$^{-2}$ s$^{-1}$ (a molar flux density of 0.49 μmol m$^{-2}$ s$^{-1}$) in the upward direction, requiring that
downward $CO_2$ diffusion be 109.5 μg m$^{-2}$ s$^{-1}$ in order to yield 88 μg m$^{-2}$ s$^{-1}$ of net surface uptake; if not
accounting for the non-diffusive resistance to net transport, the $CO_2$ diffusivity would be underestimated





by ca. 20%. The case of $CO_2$ uptake is not analogous to any pool/tube scenario in Figure 1. However,
different conditions with equal evaporation ($E = 36$ mg m$^{-2}$ s$^{-1}$) and $CO_2$ *emission* in the amount of 21.5
µg m$^{-2}$ s$^{-1}$ (by respiration, for example) would correspond to the case of zero $CO_2$ diffusion (as at the
instant $t_{eq}$), since the $CO_2$ mass fractions of both the atmosphere and the gas mixture emitted by the
surface are identical. Viewed in the traditional diffusion-only paradigm, such a situation involving a net
flux but no gradient ($F_3 = F_{3,non}$) would require a physically absurd infinite diffusivity. At this same
evaporation rate, but with lower $CO_2$ emission, diffusion of $CO_2$ would be downward, towards the surface
which is a source of $CO_2$ but a sink of the $CO_2$ mass fraction (analogous to salinity in case scenario 1 of
Section 2.2 at some instant between $t_{eq}$ and $t_f$ when the fluid emitted to the pool has a diluting effect).
Whatever the direction of net $CO_2$ transport, these case examples demonstrate the need for sometimes
substantial rectifications to flux-gradient relationships – whether expressed as a conductance, resistance,
deposition velocity, or eddy diffusivity (K-theory), – when correctly accounting for non-diffusive
transport.
**4 Discussion**
Relevant transport of a non-diffusive nature implies the need to revise the basis of flux-gradient theory,
both in the boundary layer and also at smaller scales regarding gas transfer through plant pores. One of
the key goals of micrometeorology has been the derivation of the vertical transports of mass, heat, and
momentum from profiles of wind speeds and scalar variables in the boundary layer (Businger et al.,
1971). The analyses above elucidate how gradients relate to only the diffusive components of such
exchanges. Therefore, non-diffusive flux components must be subtracted out in order to characterize
turbulent transport in terms of eddy diffusivities, a key goal of Monin-Obukhov Similarity Theory
(Obukhov, 1971).  Perhaps more important is the need to distinguish between non-diffusive and diffusive
transport mechanisms prior to assessing molecular diffusivities (conductances), as has been neglected by
the discipline of plant physiology, or ecophysiology.

When Eq. (3) is applied at the stomatal apertures where virtually all plant gas exchanges occur, it is
revealed that jets of air escape from these pores during transpiration. In the context of the scale analysis
begun in Section 3.2, it is appropriate to note that even fully open stomata occupy just 1% of leaf area
(Jones, 1983), leaving 99% cuticular and inert with regard to vital gas exchanges ($\sigma = 0.01$). As noted in
Section 3.2, this means that for the microscopic scale (D; Section 3.1) of the stomatal aperture, both the
local evaporative flux density ($E$) and therefore the lower boundary condition for the vertical velocity
($[w]_0$) predicted by Eq. (4) are two orders of magnitude greater than the 31 µm s$^{-1}$ estimated above. In
other words, a typical average airspeed exiting a stomatal aperture is 3.1 mm s$^{-1}$. For non-turbulent flow
through a cylindrical tube/aperture (i.e., Poiseuille flow), the velocity at the core of such an air current is
twice as large. If a characteristic time scale is defined for air blowing through stomata as the ratio of a
typical stomatal aperture diameter (ca. 6 µm) to this core velocity, it is found to be of order 10 ms,
illustrating that air is expelled from plants in the form of "stomatal jets".  Non-diffusive gas transport by





such airflow exiting stomata – assisting with water vapour egress but inhibiting $CO_2$ ingress – has been
previously conceived, but broadly neglected in the field of ecophysiology.

The concept of net motion and consequent non-diffusive transport out of stomata is not new, but has been
disregarded by plant ecologists. Parkinson and Penman (1970) put forth that the massive water vapour
flux from transpiration implies an outbound air current as a background against which diffusion operates.
Regrettably, however, their interpretation has largely been forgotten, having been refuted in an analysis
(Jarman, 1974) that incorrectly assumed "no net flow of air" – disregarding conservation of momentum –
and yet seems to have gained acceptance among plant physiologists (von Caemmerer and Farquhar,
1981). Similarly, Leuning (1983) recognized the relevance of non-diffusive transport and furthermore
identified excess pressure inside the stomatal cavity as the impetus for the outward airstream (which he
termed "viscous flow"), but had little impact on the mainstream characterization of stomatal conductance.
Rather, important aspects of ecophysiology continue to hinge upon the assumption that diffusion alone
transports vital gases through plant pores, disregarding both the above-mentioned studies and more
importantly the fact that gas transport mechanisms through such apertures were accurately described by
one of the great physicists of the 19[th] century.

Because Josef Stefan helped substantially to establish the fundaments of classical physics, his name is
often mentioned in the same breath as those of Boltzmann (regarding blackbody radiation) and Maxwell
(for diffusion). However, his work in the latter regard has been broadly ignored by scientists studying gas
exchanges through plant stomata. Stefan's study of evaporation from the interior of a narrow, vertical
cylinder with vapour transport into an overlying, horizontal stream of air is of particular relevance to the
discipline of ecophysiology. He determined that this is not a problem of "static diffusion", but rather
includes an element of non-diffusive transport due to a mean velocity in the direction of the vapour flux,
induced by evaporation and now commonly known as Stefan flow. Engineers know this history, refer to
such a scenario as a Stefan tube (Lienhard and Lienhard, 2000), and routinely reckon transport by Stefan
flow in addition to that caused by diffusion. Such accounting is necessary for precise control in industrial
applications such as combustion, and is described in many chemical engineering texts. The phenomenon
of transpiration through a stoma is a reasonable proxy for a Stefan tube, the main difference being that
evaporation in the Stefan tube depletes the pool of evaporating liquid, whose surface therefore recedes
downward. By contrast, the evaporating water in the stomatal cavity is continually replenished by
vascular flow from within the plant; if anything, this reinforces the magnitude of the upward vertical air
velocity, in comparison with the Stefan tube, consistent with that derived from momentum conservation
as in Eqs. (4) and (5).

Non-diffusive transport by Stefan flow has implications for defining key physiological parameters,
greater than the percentages of $CO_2$ and water vapour transport calculated above. Plant physiologists have
postulated that stomata act to maximise the ratio of carbon gain to water loss (Cowan and Farquahar,
1977) or water use efficiency (WUE), an ecosystem trait that constrains global biogeochemical cycles
(Keenan et al., 2013). In formulating this parameter, presuming molecular diffusion to be the lone



transport mechanism, the water vapour conductance is usually taken as 1.6 times that of $CO_2$ (Beer et al.,
2009), based on the ratio of their diffusivities – the inverse of the square root of the ratio of their
molecular masses, according to Graham's law. Such an assumption underlies the very concept of stomatal
control (Jones, 1983), but neglects the role of non-diffusive transport for both gases. Net momentum
exiting stomata both expedites water vapour egress and retards $CO_2$ ingress, versus the case of static
diffusion, in each case acting to reduce the WUE. Importantly, water vapour transport by stomatal jets
depends not only on physiology but also physically on the state variable $q$, according to Eq. (8).
Consistent with the determinants of $q$, as the temperature of a (saturated) stomatal environment increases,
even for a constant stomatal aperture, the WUE is reduced, wresting some control over gas exchange rates
from the plant. Perhaps equally importantly, opposition to $CO_2$ uptake by stomatal jets also should be
considered when modelling the most fundamental of biological processes, namely photosynthesis.

Accurate modelling of primary production in plants may require a fuller description of stomatal transport
mechanisms, including non-diffusive expulsion by jets. The partial pressure of $CO_2$ inside the stomata is a
key input parameter for the classic photosynthesis model (Farquhar et al., 1980), but is never directly
measured. Rather, it must be inferred from gas exchange measurements and assumptions about the
relative conductance of water vapour and $CO_2$, as described above. The amendment of such calculations
to account for non-diffusive transport of both $CO_2$ and $H_2O$ should help to improve the accuracy of
physiological models.

As a final note regarding ecophysiology, studies of plant functioning conducted using alternative gas
environments should be interpreted with care. Stomatal responses to humidity variations have been
studied in several plant species using the $He:O_2$ gas mixture termed "helox" (Mott and Parkhurst, 1991).
In the context of conservation of linear momentum, it is relevant that the effective molecular weight of
helox is just 29% that of dry air. Under equal conditions of temperature and pressure, helox has far less
density, and so during transpiration both $[w]_0$ from Eq. (4) and the non-diffusive component of stomatal
transport from Eq. (7) are 3.5 times greater than in air. The validity of helox for characterizing natural
plant functioning is thus dubious due to its low inertia versus that of air.
**Conclusions**
Evaporation ($E$) is the dominant surface gas exchange, and forces net upward momentum in the surface
layer such that the boundary condition for the vertical velocity is is $w|_0 = \frac{E}{\rho|_0}$, where $\rho|_0$ is the air density
at the surface. This non-zero vertical velocity describes Stefan flow and implies gas exchange of a non-
diffusive nature, which must be extracted from the net transport of any gas prior to relating that gas's
resultant diffusive transport component to scalar gradients, as in Monin-Obukhov Similarity Theory. Such
correction of flux-gradient theory is of particular import for descriptions of gas exchange through plant
stomata, which should be amended to account for non-diffusive transport by "stomatal jets" that help
expel water vapour but hinder the ingress of $CO_2$



**Acknowledgements**
This work is dedicated, with fondness and great esteem, to the memory of Ray Leuning whose insights
led to substantive improvements both in this work and broadly in the science of surface gas exchanges.
Investigation into this matter was funded by Spanish national project GEISpain (CGL2014-52838-C2-1-
R). The author thanks P. Serrano-Ortiz, E. P. Sánchez-Cañete, O. Pérez-Priego, S. Chamizo, A. López-
Ballesteros, R.L. Scott, J. Pérez-Quezada and anonymous reviewers for bibliographical guidance,
comments and criticisms that helped to clarify the manuscript.






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




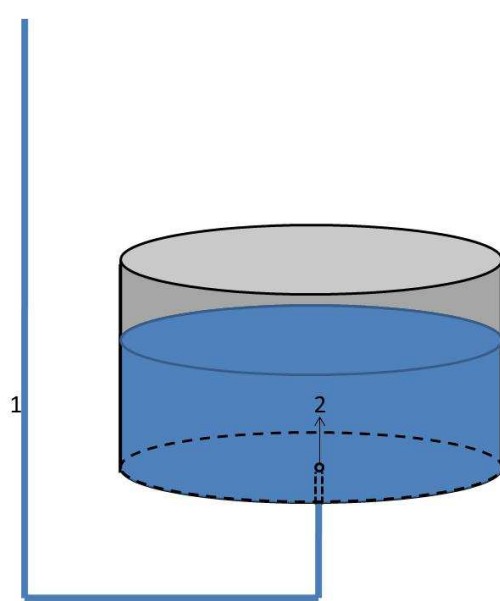

Figure 1: A pool of water being fed from below by a tube. The points indicate water (1) in the tube, and (2) in
the pool. The arrow indicates the direction of flow.

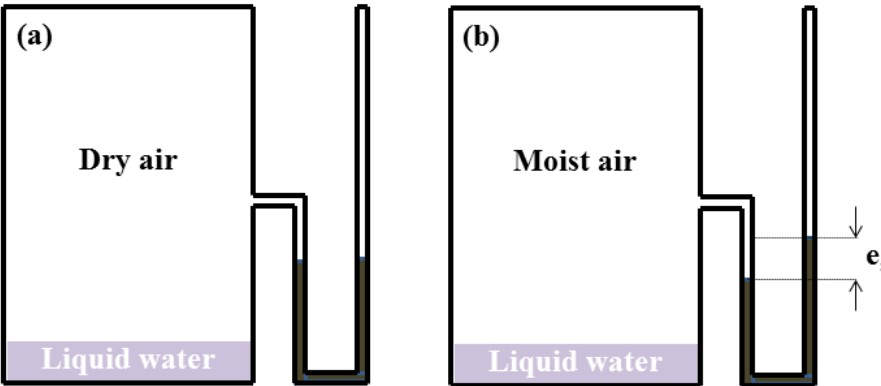

Figure 2: Illustration of evaporation incrementing air pressure. Chamber air evolves from (a) dry air initially
at atmospheric pressure; to (b) moist air at a pressure that has risen by the partial pressure of water vapour,
ultimately at equilibrium (saturation vapour pressure, $e_s$). The force generated by evaporation propels the
mercury in the manometer from its initial position.






**Table 1: List of symbols, with their meanings and units.**

| Symbol | Variable represented | S.I. Units | Tensor Order |
|---|---|---|---|
| **General Variable Representations** | | | |
| $\xi$ | An arbitrary magnitude (can represent any scalar variable) | Depends on $\xi$ | 0 (scalar) |
| $\xi_i$ | The magnitude of arbitrary variable $\xi$ for gas species $i$ | Depends on $\xi$ | 0 (scalar) |
| $\nabla\xi$ | The spatial gradient in arbitrary variable $\xi$ | Depends on $\xi$ | 1 (vector) |
| $[\xi]_0$ | The lower boundary condition for arbitrary variable $\xi$ | Depends on $\xi$ | 0 (scalar) |
| **Specific Variable Representations** | | | |
| $\Delta x, \Delta y$ | Horizontal dimensions of an analytical volume | m | 0 (scalars) |
| $\delta z$ | Vertical dimension (thickness) of an analytical volume | m | 0 (scalar) |
| $E$ | Evaporative flux density across a horizontal surface | $kg\ m^{-2}\ s^{-1}$ | 0 (component) |
| $e_s$ | Saturation vapour pressure | Pa | 0 (scalar) |
| $f$ | Mass fraction | Non-Dimensional | 0 (scalar) |
| $F_i$ | Vertical flux density of gas species $i$ | $kg\ m^{-2}\ s^{-1}$ | 0 (component) |
| $F_{i,non}$ | Non-diffusive component of $F_i$ | $kg\ m^{-2}\ s^{-1}$ | 0 (component) |
| $i$ | Index for counting gas species (as in Table 2) | - | 0 (scalar) |
| $k$ | Molecular diffusivity | $m^2\ s^{-1}$ | 0 (scalar) |
| LAI | Leaf area index | Non-Dimensional | 0 (scalar) |
| $p$ | Pressure | Pa | 0 (scalar) |
| $q$ | Specific humidity | Non-Dimensional | 0 (scalar) |
| $\rho$ | Air density | $kg\ m^{-3}$ | 0 (scalar) |
| $\sigma$ | Stomatal fraction of leaf area | Non-Dimensional | 0 (scalar) |
| $T$ | Air temperature | K | 0 (scalar) |
| $t$ | Time | s | 0 (scalar) |
| $t_0$ | Initial instant of a case scenario | s | 0 (scalar) |
| $t_{eq}$ | Equilibrium instant of a case scenario | s | 0 (scalar) |
| $t_f$ | Final instant of a case scenario | s | 0 (scalar) |
| $\boldsymbol{v}$ | Air velocity | $m\ s^{-1}$ | 1 (vector) |
| $w$ | Vertical component of $\boldsymbol{v}$ | $m\ s^{-1}$ | 0 (component) |
| WUE | Water use efficiency | Non-dimensional | 0 (scalar) |
| $z$ | Height above the surface | m | 0 (component) |





**Table 2: Gas components comprising the system to be examined, and their masses.**

| $i$ | Gas | Mass (kg) |
|---|---|---|
| 1 | Nitrogen ($N_2$) | $9.14 \cdot 10^{-16}$ |
| 2 | Oxygen ($O_2$) | $2.80 \cdot 10^{-16}$ |
| 3 | Argon ($N_2$) | $1.56 \cdot 10^{-17}$ |
| 4 | Water vapour ($H_2O$) | $1.61 \cdot 10^{-17}$ |
| 5 | Carbon dioxide ($CO_2$) | $7.36 \cdot 10^{-19}$ |
| 6 | Methane ($CH_4$) | $1.14 \cdot 10^{-21}$ |
| 7 | Nitrous oxide ($N_2O$) | $5.70 \cdot 10^{-22}$ |
| 8 | Ozone ($O_3$) | $4.01 \cdot 10^{-23}$ |



**Table 3: The first six air components by their surface exchange scale magnitude, and the net exchange of air as**
**the sum of these flux densities. Representative surface exchanges are taken from the Finnish boreal forest site**
**(Suni et al., 2003; Aaltonen et al., 2011). The $O_2$ exchange rate assumes 1:1 stoichiometry with $CO_2$.**

| gas | Typical mass flux, $F_i$ (mg m$^{-2}$ s$^{-1}$) | Corresponding molar flux (mmol m$^{-2}$ s$^{-1}$) | Source | $i$ |
|---|---|---|---|---|
| $H_2O$ | 36 | 2 | (Suni et al., 2003) | 4 |
| $CO_2$ | -0.088 | -0.002 | (Suni et al., 2003) | 5 |
| $O_2$ | 0.064 | 0.002 | (Gu, 2013) | 2 |
| $CH_4$ | -0.000032 | -0.000002 | (Aaltonen et al., 2011) | 6 |
| $O_3$ | -0.0000096 | -0.0000002 | (Suni et al., 2003) | 8 |
| $N_2O$ | 0.00000088 | 0.00000002 | (Aaltonen et al., 2011) | 7 |
| Air | 35.98 | - | This study | - |

