# Peer review of "The boundary condition for the vertical velocity and its"

_Atmospheric Chemistry and Physics, 2017_

## Referee Comment (RC1) · Anonymous Referee #1 · 19 Apr 2017

General comment The paper gives a relevant theoretical contribution in the delicate argument of the fluxes of water vapor and other gasses taking place in close vicinity of leaves and other surfaces, where evaporation takes place. This argument was overlooked in the past, leading to some improper simplifications. The paper is well written and organised. I welcome this contribution and I recommend it for publication. I suggest only some minor changes in order to make it more accessible, clear and concise. Specific indications Line 12: The vertical bars indicating processes taking place close to the surface are relatively uncommon and introduced only later in the text. This could reduce the potential readership. I recommend describing the processes by simple words in the abstract (and in the conclusions). Line 86: '. . .are the those properties. . .'

I think the term 'those' is unnecessary. The same at line 90. In equation 2 the letter k could be capital for consistency with 'K-theory' (Line 130). Line 139: What is the condition of the water present in the pool at the beginning of the experiment? Only at line 150 it is reported that the pool has zero salt mass. Line 157: The concept that the tube (or better, the liquid present in the tube) is a source of salinity is somewhat repeated. Line 170: I would recommend defining early in the text the initial conditions. The same in all case studies presented. Line 189: '…volume…at a point…', I cannot understand. A point has no volume by definition, at least in geometry. Line 223 and following. Are these four cases, all similar, strictly necessary? A single case study of the size of a leaf (e.g., 1 cm2) would simplify the text. Line 404. 'average air speed exiting a stomatal aperture is 3.1 mm s-1.' I would find interesting if the author could provide a plot or a table showing how the main physical (pore size) or environmental variables (T? P?) affect this velocity. Line 436. 'described in many chemical engineering texts'. Any references? Line 471: Any more recent references about helox experiments? Line 483: import->importance(?). Line 477 and following (Conclusions). I suggest to remove also from here uncommon symbols or to explain them all. More generally, I would still have a question: do the non-diffusive process described in the text have computational or only theoretical/descriptive effects? In Figure 2, it could be helpful if the presence of Mercury, and the circumstance that the tube is open to the atmosphere, would be indicated in the design.

---

## Referee Comment (RC2) · W. Eugster (Referee) · 20 Apr 2017

The author is known for his accurate and meticulous assessment of very fundamental aspects of atmospheric physics. In his present paper he addresses an issue that has led to many discussions before and which has not been convincingly solved so far: the magnitude of the vertical motion in the planetary boundary layer near the Earth's surface, a motion that is too small to be accurately measured with present-day state-of-the-art ultrasonic anemometers, but which is still large enough to affect (eddy covariance) flux measurements of trace gases.

So far most scientists would agree that at a certain small height above the solid ground surface, the roughness height $z|_0$ (in Kowalski's notation) the mean horizontal wind

speed must be $0\,\mathrm{m\,s^{-1}}$, and also mean vertical wind speed $w|_0$ should be $0\,\mathrm{m\,s^{-1}}$, a boundary condition that Kowalski questions on good grounds. He links $w|_0$ directly to the moisture flux density ($E$). He develops his theory based on the one-dimensional equation

$$w\rho = \sum_{i=1}^{N} w_i \rho_i \;, \tag{1}$$

with $w_i$ and $\rho_i$ being the vertical velocity and partial density of gas component $i$ in a gas mixture with $N$ components. Conceptually this is a hydrostatic approach that only allows for expansion in the vertical direction, which may exaggerate the magnitude of vertical velocity $w$. Hence, in Section 2.3 Kowalski expands to the full 3-d advection-diffusion concept that should better represent reality.

My main critique – although I must admit that my own understanding of atmospheric physics is not nearly up to the level of that of Kowalski's – is the following:

1. Kowalski primarily associates the vertical velocity at roughness height $z|_0$ with the evaporative flux $E$ but not with the vertical sensible heat flux $H$. I would assume that this is only correct for $H = 0$ W m$^{-2}$, but not for any other magnitude of sensible heat flux. In my view a partial gas density expressed as $\rho_i$ in kg m$^{-3}$ has it's volume component affected by both sensible and latent heat fluxes – and all other gas component fluxes (which however can be neglected, I agree on this aspect). An explicit treatment of the effect of $H$ would be essential in my view to help the average reader (like myself) better understand the concept and argumentation.

2. In principle the concept and analysis could be expanded to the different isotopes (stable or unstable, but the treatment of unstable isotopes would probably add yet another layer of complexity) of each gas component. At least the coverage of

stable isotopes might be helpful in context with "counter-gradient isotope fluxes" that tend to be brought up occasionally.

3.  It would be appreciated to reword some passages where plant physiologists and plant ecologists are non-neutrally qualified as partially ignorant scientists. I must admit that I had a private discussion with Graham Farquhar at a conference in Interlaken more than 10 years ago about "counter-gradient isotope fluxes" and actually had the feeling that it is fruitful in interdisciplinary work to exchange ideas between disciplines, but should not consider ourselves superior to those who start to dig into new terrain (from their perspective) – we tend to leave a similarly bleak trace if we dare to lean outside of our own territory. I think it is the strength of interdisciplinary researchers that they take the risk to be considered a non-savant outside their area of profound expertise, and we should restrain from spreading bad marks to others from other disciplines (this relates mostly to lines 394–395, 410, 415–424).

4.  The conclusions end with a very general take-home message, but since the author puts so much emphasis in his text to educate plant ecologists, it would be beneficial to have a more specific recommendation set for what plant ecologists finally are supposed to do with this new-gained knowledge. This does not explicitly become clear and the paper would benefit by having such explicit, specific recommendations that I and other could easily pik up, understand, and implement in our own calculations.

Some minor technical issues that should also be corrected:

L. 86: add "vertical" before velocity
L. 403: this appears to be the old notation of the previous (internal) version and should now read $(w|_0)$

---

## Referee Comment (RC3) · Anonymous Referee #3 · 27 Apr 2017

General comments: Vertical velocity has a tiny magnitude near surface and is difficult to measure because its magnitude is usually smaller than errors. However, vertical velocity plays a substantial role in mass and energy exchanges between land and atmosphere. For simplicity, they usually assume it is zero at surface. The author argues that it is non-zero by a "thought experiment". The author is a theoretical thinker. This paper shines light on this knowledge gap. I recommend it to be published with minor revision. Specific comments: (1) 2.1.2 The 0th Law of Thermodynamics – I do believe that this is a case from second law of thermodynamics (Postulate of Clausius, see Thermodynamics by Enrico Fermi, 1936). I don't think that "The 0th Law of Thermodynamics" is independent from second law of thermodynamics. So I suggest using

the second law of thermodynamics instead of the 0th Law so that your statements no matter heat transfer and mass diffusion are govern by the same second law of thermodynamics. Fourier's law and Fick's law are empirical relationships between fluxes and gradients. Gradients are drivers for fluxes and consequences of fluxes reduce gradients, following a single irreversible direction (entropy increasing) –equilibrium (entropy maximum) –second law of thermodynamics.

(2) Vertical velocity at surface is always positive (upward) predicted by the equation (4). Based on your thought experiment, this looks true everywhere (leaves, ground, water surface) including large scale (e.g. synoptic scale). To my knowledge, it is sure that vertical velocity is negative in high pressure system areas and positive in low pressure system areas. Therefore, it is difficult for me to understand the positive vertical velocity predicted by your theory in high pressure system areas or divergent air-flow near surface at any scale. Please clarify the conflict in your revision.

(3) Page 6 second paragraph,

It is fine to me with "vertical advection" because it is clearly defined by vertical component

It does not need to assume horizontal homogeneity.

Please also note the supplement to this comment:
http://www.atmos-chem-phys-discuss.net/acp-2017-172/acp-2017-172-RC3-supplement.pdf
* * *
General comments: Vertical velocity has a tiny magnitude near surface and is difficult to measure because its magnitude is usually smaller than errors. However, vertical velocity plays a substantial role in mass and energy exchanges between land and atmosphere. For simplicity, they usually assume it is zero at surface. The author argues that it is non-zero by a "thought experiment". The author is a theoretical thinker. This paper shines light on this knowledge gap. I recommend it to be published with minor revision.

Specific comments:

(1) 2.1.2 The $0^{th}$ Law of Thermodynamics – I do believe that this is a case from second law of thermodynamics (Postulate of Clausius, see Thermodynamics by Enrico Fermi, 1936). I don't think that "The $0^{th}$ Law of Thermodynamics" is independent from second law of thermodynamics. So I suggest using the second law of thermodynamics instead of the $0^{th}$ Law so that your statements no matter heat transfer and mass diffusion are govern by the same second law of thermodynamics. Fourier's law and Fick's law are empirical relationships between fluxes and gradients. Gradients are drivers for fluxes and consequences of fluxes reduce gradients, following a single irreversible direction (entropy increasing) – equilibrium (entropy maximum) –second law of thermodynamics.

(2) Vertical velocity at surface is always positive (upward) predicted by the equation (4). Based on your thought experiment, this looks true everywhere (leaves, ground, water surface) including large scale (e.g. synoptic scale). To my knowledge, it is sure that vertical velocity is negative in high pressure system areas and positive in low pressure system areas. Therefore, it is difficult for me to understand the positive vertical velocity predicted by your theory in high pressure system areas or divergent air-flow near surface at any scale. Please clarify the conflict in your revision.

(3) Page 6 second paragraph,

negative of the inner product $\mathbf{v} \cdot \nabla \xi$, where $\mathbf{v}$ is the fluid velocity and $\nabla$ is the gradient operator. To be clear, it can make sense to speak of "upward diffusion", but certainly not "upward advection". The tendency, in the science of surface-atmosphere exchange, to speak of "vertical advection" (e.g., Rannik et al., (2009)) is intimately related to an assumption of horizontal homogeneity, precluding horizontal scalar gradients particularly in the direction of the mean wind.

It is fine to me with "vertical advection" because it is clearly defined by vertical component

$$\mathbf{v} \cdot \nabla \xi = u \frac{\partial \xi}{\partial x} + v \frac{\partial \xi}{\partial y} + w \frac{\partial \xi}{\partial z}$$

vertical advection $w \frac{\partial \xi}{\partial z}$

horizontal advection $u \frac{\partial \xi}{\partial x} + v \frac{\partial \xi}{\partial y}$

It does not need to assume horizontal homogeneity.

[Figure]

**Fig. 1.**

---

## Author Comment (AC1) · 28 Apr 2017

The author thanks the referee for the evaluation and especially for the recommendations to improve the manuscript. In the replies that follow, the referee comments are repeated (bold font) followed by the responses from the author (normal font).

**Replies to Anonymous Referee #1**

**General comment**

**The paper gives a relevant theoretical contribution in the delicate argument of the fluxes of water vapor and other gasses taking place in close vicinity of leaves and other surfaces, where evaporation takes place. This argument was overlooked in the past, leading to some improper simplifications. The paper is well written and organised. I welcome this contribution and I recommend it for publication. I suggest only some minor changes in order to make it more accessible, clear and concise.**

The author thanks the referee for this clear endorsement.

**Specific indications**

**Line 12: The vertical bars indicating processes taking place close to the surface are relatively uncommon and introduced only later in the text. This could reduce the potential readership. I recommend describing the processes by simple words in the abstract (and in the conclusions).** The author agrees and will adopt this strategy in revising the text.

**Line 86: '. . .are the those properties. . .' I think the term 'those' is unnecessary. The same at line 90.** The author agrees and will delete these unnecessary terms.

**In equation 2 the letter k could be capital for consistency with 'K-theory' (Line 130).** The author agrees and will modify this, both in the equation and also in Table 1.

**Line 139: What is the condition of the water present in the pool at the beginning of the experiment? Only at line 150 it is reported that the pool has zero salt mass.** The condition of the water present in the pool at the beginning of the experiment varies as a function of the case scenario. In the first scenario (specified at line 150), it has zero salt mass; in the second scenario (specified at line 181), it is freshwater with salinity equal to that entering from the tube. The author agrees that the original text leaves the reader wondering for too long, and proposes to modify the sentence at line 139 to say "into the bottom of a pool (Figure 1) of salinity specified according to the two case scenarios defined below".

**Line 157: The concept that the tube (or better, the liquid present in the tube) is a source of salinity is somewhat repeated.** The author feels that such repetition is necessary in order to clarify an important point, namely that after the moment $t_{eq}$ the tube no longer represents a source of salinity, but rather dilutes the pool.

**Line 170: I would recommend defining early in the text the initial conditions. The same in all case studies presented.** This is indeed what has been done. The initial conditions of case scenarios 1 and 2 are specified early in the text (lines 148-152 for case scenario 1; lines 181-184 for case

scenario 2). Furthermore, they are presented in italic font, consistent with the words "*characteristics of the two 146 case scenarios*" at line 146. The points of view regarding (a) salinity (starting at line 154) and (b) thermodynamics (starting at line 170) refer to the same case scenario.

The author appreciates that this was less than clear, and so proposes to modify the text at lines 152-153 to say that "This first case scenario is of interest from both (a) salt/solute and (b) thermodynamic points of view:", in order that the reader clearly appreciate that points of view (a) and (b)both refer to the same conditions.

**Line 189: '. . .volume. . .at a point. . .', I cannot understand. A point has no volume by definition, at least in geometry.** Although a point is infinitesimal and has no volume in geometry, in fluid dynamics it must have a finite size, based on the continuum hypothesis. This allows mathematical specification of both state and flow properties - which only have meaning when averaging very many molecules - and their derivatives. In this context, there is a tradition of writing equations in the constant-volume (Eulerian) fluid specification to describe the fluid "at a point". In the manuscript, the quotation marks in the manuscript denote the implicit assumption regarding both the fluid as a continuous medium and the finite volume of the point under consideration.

To make this somewhat more clear to the reader, the author proposes to modify the parenthetical remark at line 189 to say '(e.g., "at a point", in an Eulerian fluid specification)'.

**Line 223 and following. Are these four cases, all similar, strictly necessary? A single case study of the size of a leaf (e.g., 1 cm2) would simplify the text.** The author feels that the four cases are indeed worth considering, for the simple reason that the derived velocities represent spatial averages that are valid over a variety of scales, the larger of which are of particular interest to meteorologists. However, the manuscript neglected to make this explicit. As a result, the author proposes to change the paragraph that began at 282 to the following:

> The representative evaporation rate prescribed in Table 3 and vertical velocity resulting from eq. (4) are valid for most of the scales defined above. Thus, the boundary condition $w|_0$ is valid for the synoptic scale (notwithstanding vertical motion aloft, such as subsidence), for the micrometeorological scale, and even for the leaf scale. In the context of scale analysis, leaves may be approximated as having equal area as the underlying surface (i.e., a unit leaf area index, or LAI=1), and equal evaporation rates as the surface in general. This latter assumption does not neglect soil evaporation, but only excludes the possibility that it dominate leaf evaporation by an order of magnitude. Thus, it will be assumed here that the assumed evaporation rate and derived vertical velocities are equally valid at synoptic (A), micrometeorological (B), and leaf (C) scales. The order of magnitude is different, however, at the microscopic scale. To show this, it will be assumed here that all leaf evaporation (or transpiration) occurs through the small fraction of the leaf that is stomatal ($\sigma$), such that both the stomatal evaporative flux density and the lower boundary condition for the vertical velocity ($w|_0$) are a factor $1/\sigma$ greater than that at larger scales. Independent of scale, Eq. (4) states that, for a positive evaporation rate, the boundary condition for the vertical velocity is non-zero and upward.

**Line 404. 'average air speed exiting a stomatal aperture is 3.1 mm s-1.' I would find interesting if the author could provide a plot or a table showing how the main physical (pore size) or environmental variables (T? P?) affect this velocity.** The roles of such variables in modulating this velocity are quite small in comparison with its near-direct, linear dependence on the evaporation

rate, as described by eq. (4). Such dependences do not, in the author's opinion, merit depiction via a plot or a table.

**Line 436. 'described in many chemical engineering texts'. Any references?** The author agrees to add references to three chemical engineering texts here (Kreith et al., 1999; Lienhard and Lienhard, 2000; Bird et al., 2002).

**Line 471: Any more recent references about helox experiments?** The author is unaware of more recent references regarding helox experiments, but points out that the Mott and Parkhurst (1991) paper has been cited more than 300 times (Web of Science).

**Line 483: import->importance(?).** The author agrees to make this proposed change.

**Line 477 and following (Conclusions).I suggest to remove also from here uncommon symbols or to explain them all. More generally, I would still have a question: do the non-diffusive process described in the text have computational or only theoretical/descriptive effects?**
The author agrees to explain the symbols in the conclusions section.

The computation of non-diffusive transport is given by equation (7). This is the amount of transport that should be subtracted from total transport in order to characterize diffusive transport, which is the quantity that is relevant to the derivation of flux-gradient relationships such as the eddy diffusivity and/or stomatal conductance. The purpose of the last three paragraphs of section 3.3 is to compute, for particular gases using representative flux magnitudes and environmental conditions, the magnitudes of these "corrections" to the eddy diffusivity and/or stomatal conductance.

**In Figure 2, it could be helpful if the presence of Mercury, and the circumstance that the tube is open to the atmosphere, would be indicated in the design.** The author agrees to annotate Figure 2 for the revised manuscript, inserting the information regarding the mercury in the manometer and the fact that the tube is open to the atmosphere.

---

## Author Response (AR1)

Dear Dr. Armin Sorooshianlaw and Natascha Töpfer,

Your e-mail of 19 May recognizes that I replied fairly early in the discussion period to all of the interactive comments, consistent with the *Atmospheric Chemistry and Physics* objective to stimulate further discussion by interested scientists. However, that e-mail also specified that "the response to the Referees shall be structured in a clear and easy to follow sequence: (1) comments from Referees, (2) author's response, (3) author's changes in manuscript". Therefore, I include here copies of my replies, with a format that orders comments from Referees (in normal font), **author's response (in bold)**, and *author's changes in manuscript (in italics)*.

Please note that the replies below are also slightly different from the on-line replies that I posted in April, and the manuscript has also changed slightly as I have had more time to improve it. All of these changes are noted below using track changes in Word, for the convenience of the editor and reviewers.

I look forward to hearing from you regarding the possibility that this manuscript can be published in *Atmospheric Chemistry and Physics*.

Sincerely,

Andrew S. Kowalski
Profesor Titular de Universidad
Departamento de Física Aplicada
Avenida Fuentenueva S/N
Universidad de Granada
18071 Granada
SPAIN
Tel: +34 958 24 90 96
Fax: +34 958 24 32 14
http://www.ugr.es/~andyk

**Changes Not Caused by Referee Comments**

The author has discovered one substantial and two minor issues with the manuscript that he believes require modifications, independent of the comments from these three referees.

The substantial issue regards a comment from the original submission, prior to publication on-line in *Atmospheric Chemistry and Physics Discussions* (comment number 6 from the 2[nd] reviewer; March, 2017), and regards the nomenclature describing eqs. (1) and (3). The reviewer's comment was very insightful, but both the author's reply and original modification to the manuscript were inappropriate.

The referee's comment was:
**I think momentum should have the unit of kg m/s, but in Eq. (3) it has kg/m2/s and thus should not be called momentum, or the equation should be expanded to show momentum. I must admit that it is not quite clear what rho is - I assumed density (kg/m3); on page 4 top, rho is defined as "fluid density" but without giving the units. If "fluid density" is in kg, then Eq. (3) would be OK, but then rho is a mass, not a density. This needs to be rectified to be consistent and understandable.**

The author's reply was:
The reviewer is correct regarding the units of momentum and density. Indeed, Eq. (3) describes momentum flux densities, and I should have been more careful with the terminology here. The references to "momentum" between equations (3) and (4) have now been changed to mention the "momentum flux density", consistent with the terminology used to describe Eq. (1).

While teaching Micrometeorology recently, the author realized a mistake he had made in this regard. The "momentum flux density", being a form of stress (like pressure), has units of kg $m^{-1}$ $s^{-2}$. By contrast, eqs. (1) and (3) have units of kg $m^{-2}$ $s^{-1}$. This represents momentum per unit volume, which might be termed a momentum density, but is better described in terms of mass flux densities, or simply flux densities of air and water vapour.

*As a result of this "old" comment from Reviewer 2 of the ACPD submission, the author has deleted the word "momentum" three times in the text: first, just prior to and describing eq. (1) (now at line 87), and then twice in the sentence following and describing eq. (3) (now at line 294).*

Minor changes:
1. *Because the text in the 4[th] paragraph of the introduction is truly supportive of the 3[rd] paragraph of the introduction, the author has removed the paragraph break and merged these two paragraphs into one.*
2. The author has deleted the repeated word "is" in the second line of the Conclusions section.

**Replies to Anonymous Referee #1**

**General comment**

**The paper gives a relevant theoretical contribution in the delicate argument of the fluxes of water vapor and other gasses taking place in close vicinity of leaves and other surfaces, where evaporation takes place. This argument was overlooked in the past, leading to some improper simplifications. The paper is well written and organised. I welcome this contribution and I recommend it for publication. I suggest only some minor changes in order to make it more accessible, clear and concise.**

The author thanks the referee for this clear endorsement. *This comment requires no change to the manuscript.*

**Specific indications**

**Line 12: The vertical bars indicating processes taking place close to the surface are relatively uncommon and introduced only later in the text. This could reduce the potential readership. I recommend describing the processes by simple words in the abstract (and in the conclusions).** The author agrees. *The author has changed the abstract accordingly, and also slightly changed conclusions where such wording was already largely in place (see reply to comment about line 477 below).*

**Line 86: '. . .are the those properties. . .' I think the term 'those' is unnecessary. The same at line 90.** The author agrees. *These unnecessary terms have been deleted.*

**In equation 2 the letter k could be capital for consistency with 'K-theory' (Line 130).** The author agrees. *The variable k has been made upper-case in the text (now at lines 117 and 136), in the equation, and also in Table 1.*

**Line 139: What is the condition of the water present in the pool at the beginning of the experiment? Only at line 150 it is reported that the pool has zero salt mass.** The condition of the water present in the pool at the beginning of the experiment varies as a function of the case scenario. In the first scenario (specified at line 150), it has zero salt mass; in the second scenario (specified at line 181), it is freshwater with salinity equal to that entering from the tube. The author agrees that the original text leaves the reader wondering for too long. *The author has modified the sentence (now at line 145) to say "into the bottom of a pool (Figure 1) of salinity specified according to the two case scenarios defined below".*

**Line 157: The concept that the tube (or better, the liquid present in the tube) is a source of salinity is somewhat repeated.** The author feels that such repetition is necessary in order to clarify an important point, namely that after the moment $t_{eq}$ the tube no longer represents a source of salinity, but rather dilutes the pool. *The author has made no changes as a result of this comment.*

**Line 170: I would recommend defining early in the text the initial conditions. The same in all case studies presented.** This is indeed what has been done. The initial conditions of case scenarios 1 and 2 are specified early in the text (lines 148-152 for case scenario 1; lines 181-184 for case scenario 2). Furthermore, they are presented in italic font, consistent with the words *"characteristics of the two 146 case scenarios"* at line 146. The points of view regarding (a) salinity (starting at line 154) and (b) thermodynamics (starting at line 170) refer to the same case scenario.

*The author appreciates that this was less than clear, and has modified the text (now at line 162) to say that "This first case scenario is of interest from both (a) salt/solute and (b) thermodynamic points of view:", in order that the reader clearly appreciate that points of view (a) and (b)both refer to the same conditions.*

**Line 189: '. . .volume. . .at a point. . .', I cannot understand. A point has no volume by definition, at least in geometry.** Although a point is infinitesimal and has no volume in geometry, in fluid dynamics it must have a finite size, based on the continuum hypothesis. This allows mathematical specification of both state and flow properties - which only have meaning when averaging very many molecules - and their derivatives. In this context, there is a tradition of writing equations in the constant-volume (Eulerian)

fluid specification to describe the fluid "at a point". In the manuscript, the quotation marks in the manuscript denote the implicit assumption regarding both the fluid as a continuous medium and the finite volume of the point under consideration.

*To make this somewhat more clear to the reader, the author has modified the parenthetical remark – now at line 200 - to say '(e.g., "at a point", in an Eulerian fluid specification)'.*

**Line 223 and following. Are these four cases, all similar, strictly necessary? A single case study of the size of a leaf (e.g., 1 cm2) would simplify the text.** The author feels that the four cases are indeed worth considering, for the simple reason that the derived velocities represent spatial averages that are valid over a variety of scales, the larger of which are of particular interest to meteorologists. However, the manuscript neglected to make this explicit. *As a result, the author has changed the paragraph that begins at 299 to the following:*

> The representative evaporation rate prescribed in Table 3 and vertical velocity resulting from eq. (4) are valid for most of the scales defined above. Thus, the boundary condition $w|_0$ is valid for the synoptic scale (notwithstanding vertical motion aloft, such as subsidence), for the micrometeorological scale, and even for the leaf scale. In the context of scale analysis, leaves may be approximated as having equal area as the underlying surface (i.e., a unit leaf area index, or LAI=1), and equal evaporation rates as the surface in general. This latter assumption does not neglect soil evaporation, but only excludes the possibility that it dominate leaf evaporation by an order of magnitude. Thus, it will be assumed here that the assumed evaporation rate and derived vertical velocities are equally valid at synoptic (A), micrometeorological (B), and leaf (C) scales. The order of magnitude is different, however, at the microscopic (D) scale. To show this, it will be assumed here that all leaf evaporation (or transpiration) occurs through the small fraction of the leaf that is stomatal ($\sigma$), such that both the stomatal evaporative flux density and the lower boundary condition for the vertical velocity ($w|_0$) are a factor $1/\sigma$ greater than that at larger scales. Independent of scale, Eq. (4) states that, for a positive evaporation rate, the boundary condition for the vertical velocity is non-zero and upward.

**Line 404. 'average air speed exiting a stomatal aperture is 3.1 mm s-1.' I would find interesting if the author could provide a plot or a table showing how the main physical (pore size) or environmental variables (T? P?) affect this velocity.** The roles of such variables in modulating this velocity are quite small in comparison with its near-direct, linear dependence on the evaporation rate, as described by eq. (4). Such dependences do not, in the author's opinion, merit depiction via a plot or a table. *The author has made no changes as a result of this comment.*

**Line 436. 'described in many chemical engineering texts'. Any references?** The author agrees. *The author has added references to three chemical engineering texts here (Kreith et al., 1999; Lienhard and Lienhard, 2000; Bird et al., 2002).*

**Line 471: Any more recent references about helox experiments?** The author is unaware of more recent references regarding helox experiments, but points out that the Mott and Parkhurst (1991) paper has been cited more than 300 times (Web of Science). *The author has made no changes as a result of this comment.*

**Line 483: import->importance(?).** *The author has made this proposed change.*

**Line 477 and following (Conclusions).I suggest to remove also from here uncommon symbols or to explain them all. More generally, I would still have a question: do the non-diffusive process described in the text have computational or only theoretical/descriptive effects?**
The author agreed to explain the symbols in the conclusions section. However, upon revising the conclusions, the author finds that the "uncommon" symbols are already largely explained. The sentence in question states that "the boundary condition for the vertical velocity is $w|_0 = \frac{E}{\rho|_0}$, where $\rho|_0$ is the air density at the surface". *As a result of this comment, the author has added at line 510 the adjective "lower" such that this sentence now reads* "the lower boundary condition for the vertical velocity is $w|_0 = \frac{E}{\rho|_0}$, where $\rho|_0$ is the air density at the surface". In this way, both of the "uncommon" symbols are explained.

The computation of non-diffusive transport is given by equation (7). This is the amount of transport that should be subtracted from total transport in order to characterize diffusive transport, which is the quantity that is relevant to the derivation of flux-gradient relationships such as the eddy diffusivity and/or stomatal conductance. The purpose of the last three paragraphs of section 3.3 is to compute, for particular gases using representative flux magnitudes and environmental conditions, the magnitudes of these "corrections" to the eddy diffusivity and/or stomatal conductance. *The author has made no changes as a result of this comment about computational/theoretical effects.*

**In Figure 2, it could be helpful if the presence of Mercury, and the circumstance that the tube is open to the atmosphere, would be indicated in the design.** The author agrees. *The author has modified Figure 2, inserting the information regarding the mercury in the manometer and the fact that the tube is open to the atmosphere.*

**Replies to Referee #2 (W. Eugster)**

**The author is known for his accurate and meticulous assessment of very fundamental aspects of atmospheric physics. In his present paper he addresses an issue that has led to many discussions before and which has not been convincingly solved so far: the magnitude of the vertical motion in the planetary boundary layer near the Earth's surface, a motion that is too small to be accurately measured with present-day state-of-the-art ultrasonic anemometers, but which is still large enough to affect (eddy covariance) flux measurements of trace gases.**

**So far most scientists would agree that at a certain small height above the solid ground surface, the roughness height $z\big|_0$ (in Kowalski's notation) the mean horizontal wind speed must be 0 ms$^{-1}$, and also mean vertical wind speed $w\big|_0$ should be 0 ms$^{-1}$, a boundary condition that Kowalski questions on good grounds. He links $w\big|_0$ directly to the moisture flux density (E). He develops his theory based on the one-dimensional equation**

$$w\rho = \sum_{i=1}^{N} w_i \rho_i. \tag{1}$$

**with $w_i$ and $\rho_i$ being the vertical velocity and partial density of gas component *i* in a gas mixture with *N* components. Conceptually this is a hydrostatic approach that only allows for expansion in the vertical direction, which may exaggerate the magnitude of vertical velocity w. Hence, in Section 2.3 Kowalski expands to the full 3-d advectiondiffusion concept that should better represent reality.**

The author thanks Dr. Eugster for this assessment, which shows that the manuscript has managed to communicate the essence of the theory being developed. *The author has made no changes as a result of this comment.*

**My main critique – although I must admit that my own understanding of atmospheric physics is not nearly up to the level of that of Kowalski's – is the following:**

**1. Kowalski primarily associates the vertical velocity at roughness height $z\big|_0$ with the evaporative flux E but not with the vertical sensible heat flux H. I would assume that this is only correct for H = 0 W m$^{-2}$, but not for any other magnitude of sensible heat flux. In my view a partial gas density expressed as $\rho_i$ in kg m$^{-3}$ has it's volume component affected by both sensible and latent heat fluxes – and all other gas component fluxes (which however can be neglected, I agree on this aspect). An explicit treatment of the effect of H would be essential in my view to help the average reader (like myself) better understand the concept and argumentation.**

The author disagrees here, and points to his previous publications that address this very issue.

Certainly, it is traditional in micrometeorology to infer a mean velocity in the direction of the sensible heat flux (Webb et al., 1980; "WPL"), as Dr. Eugster asserts. However, Kowalski (2012) showed that this inference is an artefact of imprecise averaging procedures. Perhaps the simplest illustration of the WPL error is given by the scenario visualized in the artless drawing in Figure X below, where turbulent air convection is enclosed within a stationary chamber with an upward heat flux under steady-state conditions.

[Figure]

*Figure X. A chamber with a heated floor (red), chilled ceiling (blue), and insulated side walls (grey). The inside air is in steady state and turbulent (convective) due to an unstable lapse rate.*

The most fundamental definition of the mean velocity of a system is the ratio of its displacement to the time elapsed. For steady-state air confined to a stationary chamber, its displacement over long time periods is clearly zero, and therefore so is the average velocity, notwithstanding WPL predictions of an upward velocity associated with the upward heat flux.

For non-steady-state conditions, average velocities can occur due to expansion/compression of the air layer near a surface boundary. For example, in Figure X, assuming that temperature effects dominate those of pressure in determining air density, there is more air near the cool ceiling than at the hot floor. Thus, eliminating floor heating and ceiling cooling would result in a net downward transport of air. Such effects are treated by Kowalski and Serrano-Ortiz (2008) for a simple scenario, neglecting evaporation's influence on the average velocity, and demonstrating conceptual errors in the WPL velocity. A more general derivation, combining the effects of compressibility and vapor exchanges, is beyond the scope of the current discussion.

*The author has made no changes as a result of this comment.*

**2. In principle the concept and analysis could be expanded to the different isotopes (stable or unstable, but the treatment of unstable isotopes would probably add yet another layer of complexity) of each gas component. At least the coverage of stable isotopes might be helpful in context with "counter-gradient isotope fluxes" that tend to be brought up occasionally.**

The author agrees that the concept and analysis could be expanded to the issue of isotope transport, but is ignorant of the phenomenon of "counter-gradient isotope fluxes". Perhaps Dr. Eugster could provide references for these observations (?). Otherwise, it seems that this issue is beyond the scope of the present manuscript, but could be worthy of future investigation.

*The author has made no changes as a result of this comment, but remains interested in learning more about "counter-gradient isotope fluxes".*

**3. It would be appreciated to reword some passages where plant physiologists and plant ecologists are non-neutrally qualified as partially ignorant scientists. I must admit that I had a private discussion with Graham Farquhar at a conference in Interlaken more than 10 years ago about "counter-gradient isotope fluxes" and actually had the feeling that it is fruitful in interdisciplinary work to exchange ideas between disciplines, but should not consider ourselves superior to those who start to dig into new terrain (from their perspective) – we tend to leave a similarly bleak trace if we dare to lean outside of our own territory. I think it is the strength of interdisciplinary researchers that they take the risk to be considered a non-savant outside their area of profound expertise, and we should restrain from spreading bad marks to others from other disciplines (this relates mostly to lines 394–395, 410, 415–424).**

The author is aware that some readers find his writing style to be offensive, has worked hard to try to correct this problem, and is not surprised to find that it persists. There can be little doubt regarding the benefits of exchanging ideas between disciplines, and any specific recommendations would be very welcome regarding how to reword passages so as to be more neutral.

However, having said that, the author is unwilling to refrain from criticizing theories or procedures that are incorrect. The objective here is not to consider anyone as superior, but rather to discover and defend the truth, and the author has taken great care to do this based solidly on fundamental physical laws. From the author's point of view, Jarman (1974) neglected momentum conservation, erroneously classified the description of Parkinson and Penman (1970) as "incorrect" (twice) or "substantially incorrect" (twice more), and misled an entire community of scientists along a mistaken path for several decades. The key question then is how correct this error and prevent its further propagation.

*In an attempt to comply with Dr. Eugster's suggestion and avoid defaming a particular discipline, the author has adopted the following changes*

- *Lines 419: delete ",as has been neglected by the discipline of plant physiology, or ecophysiology";*
- *Line 434: delete ", but broadly neglected in the field of ecophysiology"*
- *Line 447: delete "among plant physiologists"*

**4. The conclusions end with a very general take-home message, but since the author puts so much emphasis in his text to educate plant ecologists, it would be beneficial to have a more specific recommendation set for what plant ecologists finally are supposed to do with this new-gained knowledge. This does not explicitly become clear and the paper would benefit by having such explicit, specific recommendations that I and other could easily pik up, understand, and implement in our own calculations.**

Unfortunately, the author has not been able to develop a "quick fix": an algorithm or equation that would immediately correct WUE or $c_i$ by accounting for non-diffusive transport. This therefore falls under the category of future research, and is open to any scientist who may have better ideas about solving these tricky issues. *The author has made no changes as a result of this comment.*

**"Minor technical issues":**
**L. 86: add "vertical" before velocity**

*The author has made this change (now line 89).*

**L. 403: this appears to be the old notation of the previous (internal) version and should now read (w $|_0$)**

*The author has made this change (now line 427). The author found and corrected the same problem in the last paragraph of the discussion (now line 505).*

**Replies to Anonymous Referee #3**

**General comments: Vertical velocity has a tiny magnitude near surface and is difficult to measure because its magnitude is usually smaller than errors. However, vertical velocity plays a substantial role in mass and energy exchanges between land and atmosphere. For simplicity, they usually assume it is zero at surface. The author argues that it is non-zero by a "thought experiment". The author is a theoretical thinker. This paper shines light on this knowledge gap. I recommend it to be published with minor revision.**

The author thanks the referee for this endorsement. The *author has made no changes as a result of this comment.*

**Specific comments:**

**(1) 2.1.2 The 0th Law of Thermodynamics – I do believe that this is a case from second law of thermodynamics (Postulate of Clausius, see Thermodynamics by Enrico Fermi, 1936). I don't think that "The 0th Law of Thermodynamics" is independent from second law of thermodynamics. So I suggest using the second law of thermodynamics instead of the 0th Law so that your statements no matter heat transfer and mass diffusion are govern by the same second law of thermodynamics. Fourier's law and Fick's law are empirical relationships between fluxes and gradients. Gradients are drivers for fluxes and consequences of fluxes reduce gradients, following a single irreversible direction (entropy increasing) – equilibrium (entropy maximum) –second law of thermodynamics.**

The 1936 textbook cited, although authored by a great physicist, is nonetheless out of date regarding this issue. Modern physics texts (e.g., Giancoli, 1984) recognize that the $1^{st}$ and $2^{nd}$ Laws, although definitively stated first, logically depend on the prior assertion of the $0^{th}$ Law; this explains its odd name. Formalization of the $0^{th}$ Law occurred in the mid-1930s, but was not broadly accepted until well after the publication of Fermi's textbook. The *author has made no changes as a result of this comment.*

**(2) Vertical velocity at surface is always positive (upward) predicted by the equation (4). Based on your thought experiment, this looks true everywhere (leaves, ground, water surface) including large scale (e.g. synoptic scale). To my knowledge, it is sure that vertical velocity is negative in high pressure system areas and positive in low pressure system areas. Therefore, it is difficult for me to understand the positive vertical velocity predicted by your theory in high pressure system areas or divergent air-flow near surface at any scale. Please clarify the conflict in your revision.**

Synoptic-scale velocities are of order 3 cm s$^{-1}$ (e.g., Carlson and Stull, 1986, Subsidence in the nocturnal boundary layer, *J. Clim. Appl. Met.*, **25**, 1088-1099). Whether the boundary condition at the surface is the traditionally conceived $\mathbf{w}\big|_0 = 0$ cm s$^{-1}$, or 0.000031 cm s$^{-1}$ as derived from eq. (4), synoptic-scale subsidence implies a convergence in the vertical winds between the surface and the height at which it occurs. It seems that there is no conflict that requires clarification. The *author has made no changes as a result of this comment.*

**(3) Page 6 second paragraph, It is fine to me with "vertical advection" because it is clearly defined by vertical component It does not need to assume horizontal homogeneity.**

The author agrees, and initially proposed simply to delete the last two sentences of this paragraph. However, upon careful reconsideration, the author wishes to finish this paragraph with a statement to clarify a key distinction between advection and diffusion. *Therefore, the author has deleted the two sentences at the end of the first paragraph of section 2.3 (and consequentially the Rannik et al., 2009 paper has been removed from the references section), and added a new sentence to finish this paragraph (now at lines 213-214), to say that:* "Thus advection, unlike diffusion, is not a form of transport, but rather a consequence of differential transport."

Dear Dr. Armin Sorooshianlaw and Natascha Töpfer,

Your e-mail of 19 May recognizes that I replied fairly early in the discussion period to all of the interactive comments, consistent with the *Atmospheric Chemistry and Physics* objective to stimulate further discussion by interested scientists. However, that e-mail also specified that "the response to the Referees shall be structured in a clear and easy to follow sequence: (1) comments from Referees, (2) author's response, (3) author's changes in manuscript". Therefore, I include here copies of my replies, with a format that orders comments from Referees (in normal font), **author's response (in bold)**, and *author's changes in manuscript (in italics)*.

Please note that the replies below are also slightly different from the on-line replies that I posted in April, and the manuscript has also changed slightly as I have had more time to improve it. All of these changes are noted below using track changes in Word, for the convenience of the editor and reviewers.

I look forward to hearing from you regarding the possibility that this manuscript can be published in *Atmospheric Chemistry and Physics*.

Sincerely,

Andrew S. Kowalski
Profesor Titular de Universidad
Departamento de Física Aplicada
Avenida Fuentenueva S/N
Universidad de Granada
18071 Granada
SPAIN
Tel: +34 958 24 90 96
Fax: +34 958 24 32 14
http://www.ugr.es/~andyk

**Changes Not Caused by Referee Comments**

The author has discovered one substantial and two minor issues with the manuscript that he believes require modifications, independent of the comments from these three referees.

The substantial issue regards a comment from the original submission, prior to publication on-line in *Atmospheric Chemistry and Physics Discussions* (comment number 6 from the 2[nd] reviewer; March, 2017), and regards the nomenclature describing eqs. (1) and (3). The reviewer's comment was very insightful, but both the author's reply and original modification to the manuscript were inappropriate.

The referee's comment was:
**I think momentum should have the unit of kg m/s, but in Eq. (3) it has kg/m2/s and thus should not be called momentum, or the equation should be expanded to show momentum. I must admit that it is not quite clear what rho is - I assumed density (kg/m3); on page 4 top, rho is defined as "fluid density" but without giving the units. If "fluid density" is in kg, then Eq. (3) would be OK, but then rho is a mass, not a density. This needs to be rectified to be consistent and understandable.**

The author's reply was:
The reviewer is correct regarding the units of momentum and density. Indeed, Eq. (3) describes momentum flux densities, and I should have been more careful with the terminology here. The references to "momentum" between equations (3) and (4) have now been changed to mention the "momentum flux density", consistent with the terminology used to describe Eq. (1).

While teaching Micrometeorology recently, the author realized a mistake he had made in this regard. The "momentum flux density", being a form of stress (like pressure), has units of $kg\ m^{-1}\ s^{-2}$. By contrast, eqs. (1) and (3) have units of $kg\ m^{-2}\ s^{-1}$. This represents momentum per unit volume, which might be termed a momentum density, but is better described in terms of mass flux densities, or simply flux densities of air and water vapour.

*As a result of this "old" comment from Reviewer 2 of the ACPD submission, the author has deleted the word "momentum" three times in the text: first, just prior to and describing eq. (1) (now at line 87), and then twice in the sentence following and describing eq. (3) (now at line 294).*

Minor changes:
1. *Because the text in the 4[th] paragraph of the introduction is truly supportive of the 3[rd] paragraph of the introduction, the author has removed the paragraph break and merged these two paragraphs into one.*
2. The author has deleted the repeated word "is" in the second line of the Conclusions section.

**Replies to Anonymous Referee #1**

**General comment**

**The paper gives a relevant theoretical contribution in the delicate argument of the fluxes of water vapor and other gasses taking place in close vicinity of leaves and other surfaces, where evaporation takes place. This argument was overlooked in the past, leading to some improper simplifications. The paper is well written and organised. I welcome this contribution and I recommend it for publication. I suggest only some minor changes in order to make it more accessible, clear and concise.**

The author thanks the referee for this clear endorsement. *This comment requires no change to the manuscript.*

**Specific indications**

**Line 12: The vertical bars indicating processes taking place close to the surface are relatively uncommon and introduced only later in the text. This could reduce the potential readership. I recommend describing the processes by simple words in the abstract (and in the conclusions).** The author agrees. *The author has changed the abstract accordingly, and also slightly changed conclusions where such wording was already largely in place (see reply to comment about line 477 below).*

**Line 86: '. . .are the those properties. . .' I think the term 'those' is unnecessary. The same at line 90.** The author agrees. *These unnecessary terms have been deleted.*

**In equation 2 the letter k could be capital for consistency with 'K-theory' (Line 130).** The author agrees. *The variable k has been made upper-case in the text (now at lines 117 and 136), in the equation, and also in Table 1.*

**Line 139: What is the condition of the water present in the pool at the beginning of the experiment? Only at line 150 it is reported that the pool has zero salt mass.** The condition of the water present in the pool at the beginning of the experiment varies as a function of the case scenario. In the first scenario (specified at line 150), it has zero salt mass; in the second scenario (specified at line 181), it is freshwater with salinity equal to that entering from the tube. The author agrees that the original text leaves the reader wondering for too long. *The author has modified the sentence (now at line 145) to say "into the bottom of a pool (Figure 1) of salinity specified according to the two case scenarios defined below".*

**Line 157: The concept that the tube (or better, the liquid present in the tube) is a source of salinity is somewhat repeated.** The author feels that such repetition is necessary in order to clarify an important point, namely that after the moment $t_{eq}$ the tube no longer represents a source of salinity, but rather dilutes the pool. *The author has made no changes as a result of this comment.*

**Line 170: I would recommend defining early in the text the initial conditions. The same in all case studies presented.** This is indeed what has been done. The initial conditions of case scenarios 1 and 2 are specified early in the text (lines 148-152 for case scenario 1; lines 181-184 for case scenario 2). Furthermore, they are presented in italic font, consistent with the words "*characteristics of the two 146 case scenarios*" at line 146. The points of view regarding (a) salinity (starting at line 154) and (b) thermodynamics (starting at line 170) refer to the same case scenario.

*The author appreciates that this was less than clear, and has modified the text (now at line 162) to say that "This first case scenario is of interest from both (a) salt/solute and (b) thermodynamic points of view:", in order that the reader clearly appreciate that points of view (a) and (b)both refer to the same conditions.*

**Line 189: '. . .volume. . .at a point. . .', I cannot understand. A point has no volume by definition, at least in geometry.** Although a point is infinitesimal and has no volume in geometry, in fluid dynamics it must have a finite size, based on the continuum hypothesis. This allows mathematical specification of both state and flow properties - which only have meaning when averaging very many molecules - and their derivatives. In this context, there is a tradition of writing equations in the constant-volume (Eulerian)

fluid specification to describe the fluid "at a point". In the manuscript, the quotation marks in the manuscript denote the implicit assumption regarding both the fluid as a continuous medium and the finite volume of the point under consideration.

*To make this somewhat more clear to the reader, the author has modified the parenthetical remark – now at line 200 - to say '(e.g., "at a point", in an Eulerian fluid specification)'.*

**Line 223 and following. Are these four cases, all similar, strictly necessary? A single case study of the size of a leaf (e.g., 1 cm2) would simplify the text.** The author feels that the four cases are indeed worth considering, for the simple reason that the derived velocities represent spatial averages that are valid over a variety of scales, the larger of which are of particular interest to meteorologists. However, the manuscript neglected to make this explicit. *As a result, the author has changed the paragraph that begins at 299 to the following:*

> The representative evaporation rate prescribed in Table 3 and vertical velocity resulting from eq. (4) are valid for most of the scales defined above. Thus, the boundary condition $w\big|_0$ is valid for the synoptic scale (notwithstanding vertical motion aloft, such as subsidence), for the micrometeorological scale, and even for the leaf scale. In the context of scale analysis, leaves may be approximated as having equal area as the underlying surface (i.e., a unit leaf area index, or LAI=1), and equal evaporation rates as the surface in general. This latter assumption does not neglect soil evaporation, but only excludes the possibility that it dominate leaf evaporation by an order of magnitude. Thus, it will be assumed here that the assumed evaporation rate and derived vertical velocities are equally valid at synoptic (A), micrometeorological (B), and leaf (C) scales. The order of magnitude is different, however, at the microscopic (D) scale. To show this, it will be assumed here that all leaf evaporation (or transpiration) occurs through the small fraction of the leaf that is stomatal ($\sigma$), such that both the stomatal evaporative flux density and the lower boundary condition for the vertical velocity ($w\big|_0$) are a factor $1/\sigma$ greater than that at larger scales. Independent of scale, Eq. (4) states that, for a positive evaporation rate, the boundary condition for the vertical velocity is non-zero and upward.

**Line 404. 'average air speed exiting a stomatal aperture is 3.1 mm s-1.' I would find interesting if the author could provide a plot or a table showing how the main physical (pore size) or environmental variables (T? P?) affect this velocity.** The roles of such variables in modulating this velocity are quite small in comparison with its near-direct, linear dependence on the evaporation rate, as described by eq. (4). Such dependences do not, in the author's opinion, merit depiction via a plot or a table. *The author has made no changes as a result of this comment.*

**Line 436. 'described in many chemical engineering texts'. Any references?** The author agrees. *The author has added references to three chemical engineering texts here (Kreith et al., 1999; Lienhard and Lienhard, 2000; Bird et al., 2002).*

**Line 471: Any more recent references about helox experiments?** The author is unaware of more recent references regarding helox experiments, but points out that the Mott and Parkhurst (1991) paper has been cited more than 300 times (Web of Science). *The author has made no changes as a result of this comment.*

**Line 483: import->importance(?).** *The author has made this proposed change.*

**Line 477 and following (Conclusions).I suggest to remove also from here uncommon symbols or to explain them all. More generally, I would still have a question: do the non-diffusive process described in the text have computational or only theoretical/descriptive effects?**
The author agreed to explain the symbols in the conclusions section. However, upon revising the conclusions, the author finds that the "uncommon" symbols are already largely explained. The sentence in question states that "the boundary condition for the vertical velocity is $w\big|_0 = \frac{E}{\rho|_0}$, where $\rho\big|_0$ is the air density at the surface". *As a result of this comment, the author has added at line 510 the adjective "lower" such that this sentence now reads* "the lower boundary condition for the vertical velocity is $w\big|_0 = \frac{E}{\rho|_0}$, where $\rho\big|_0$ is the air density at the surface". In this way, both of the "uncommon" symbols are explained.

The computation of non-diffusive transport is given by equation (7). This is the amount of transport that should be subtracted from total transport in order to characterize diffusive transport, which is the quantity that is relevant to the derivation of flux-gradient relationships such as the eddy diffusivity and/or stomatal conductance. The purpose of the last three paragraphs of section 3.3 is to compute, for particular gases using representative flux magnitudes and environmental conditions, the magnitudes of these "corrections" to the eddy diffusivity and/or stomatal conductance. *The author has made no changes as a result of this comment about computational/theoretical effects.*

**In Figure 2, it could be helpful if the presence of Mercury, and the circumstance that the tube is open to the atmosphere, would be indicated in the design.** The author agrees. *The author has modified Figure 2, inserting the information regarding the mercury in the manometer and the fact that the tube is open to the atmosphere.*

**Replies to Referee #2 (W. Eugster)**

**The author is known for his accurate and meticulous assessment of very fundamental aspects of atmospheric physics. In his present paper he addresses an issue that has led to many discussions before and which has not been convincingly solved so far: the magnitude of the vertical motion in the planetary boundary layer near the Earth's surface, a motion that is too small to be accurately measured with present-day state-of-the-art ultrasonic anemometers, but which is still large enough to affect (eddy covariance) flux measurements of trace gases.**

**So far most scientists would agree that at a certain small height above the solid ground surface, the roughness height $z|_0$ (in Kowalski's notation) the mean horizontal wind speed must be 0 ms$^{-1}$, and also mean vertical wind speed $w|_0$ should be 0 ms$^{-1}$, a boundary condition that Kowalski questions on good grounds. He links $w|_0$ directly to the moisture flux density (E). He develops his theory based on the one-dimensional equation**

$$w\rho = \sum_{i=1}^{N} w_i \rho_i. \qquad (1)$$

**with $w_i$ and $\rho_i$ being the vertical velocity and partial density of gas component *i* in a gas mixture with *N* components. Conceptually this is a hydrostatic approach that only allows for expansion in the vertical direction, which may exaggerate the magnitude of vertical velocity w. Hence, in Section 2.3 Kowalski expands to the full 3-d advectiondiffusion concept that should better represent reality.**

The author thanks Dr. Eugster for this assessment, which shows that the manuscript has managed to communicate the essence of the theory being developed. *The author has made no changes as a result of this comment.*

**My main critique – although I must admit that my own understanding of atmospheric physics is not nearly up to the level of that of Kowalski's – is the following:**

**1. Kowalski primarily associates the vertical velocity at roughness height $z|_0$ with the evaporative flux E but not with the vertical sensible heat flux H. I would assume that this is only correct for H = 0 W m$^{-2}$, but not for any other magnitude of sensible heat flux. In my view a partial gas density expressed as $\rho_i$ in kg m$^{-3}$ has it's volume component affected by both sensible and latent heat fluxes – and all other gas component fluxes (which however can be neglected, I agree on this aspect). An explicit treatment of the effect of H would be essential in my view to help the average reader (like myself) better understand the concept and argumentation.**

The author disagrees here, and points to his previous publications that address this very issue.

Certainly, it is traditional in micrometeorology to infer a mean velocity in the direction of the sensible heat flux (Webb et al., 1980; "WPL"), as Dr. Eugster asserts. However, Kowalski (2012) showed that this inference is an artefact of imprecise averaging procedures. Perhaps the simplest illustration of the WPL error is given by the scenario visualized in the artless drawing in Figure X below, where turbulent air convection is enclosed within a stationary chamber with an upward heat flux under steady-state conditions.

[Figure]

*Figure X. A chamber with a heated floor (red), chilled ceiling (blue), and insulated side walls (grey). The inside air is in steady state and turbulent (convective) due to an unstable lapse rate.*

The most fundamental definition of the mean velocity of a system is the ratio of its displacement to the time elapsed. For steady-state air confined to a stationary chamber, its displacement over long time periods is clearly zero, and therefore so is the average velocity, notwithstanding WPL predictions of an upward velocity associated with the upward heat flux.

For non-steady-state conditions, average velocities can occur due to expansion/compression of the air layer near a surface boundary. For example, in Figure X, assuming that temperature effects dominate those of pressure in determining air density, there is more air near the cool ceiling than at the hot floor. Thus, eliminating floor heating and ceiling cooling would result in a net downward transport of air. Such effects are treated by Kowalski and Serrano-Ortiz (2008) for a simple scenario, neglecting evaporation's influence on the average velocity, and demonstrating conceptual errors in the WPL velocity. A more general derivation, combining the effects of compressibility and vapor exchanges, is beyond the scope of the current discussion.

*The author has made no changes as a result of this comment.*

**2. In principle the concept and analysis could be expanded to the different isotopes (stable or unstable, but the treatment of unstable isotopes would probably add yet another layer of complexity) of each gas component. At least the coverage of stable isotopes might be helpful in context with "counter-gradient isotope fluxes" that tend to be brought up occasionally.**

The author agrees that the concept and analysis could be expanded to the issue of isotope transport, but is ignorant of the phenomenon of "counter-gradient isotope fluxes". Perhaps Dr. Eugster could provide references for these observations (?). Otherwise, it seems that this issue is beyond the scope of the present manuscript, but could be worthy of future investigation.

*The author has made no changes as a result of this comment, but remains interested in learning more about "counter-gradient isotope fluxes".*

**3. It would be appreciated to reword some passages where plant physiologists and plant ecologists are non-neutrally qualified as partially ignorant scientists. I must admit that I had a private discussion with Graham Farquhar at a conference in Interlaken more than 10 years ago about "counter-gradient isotope fluxes" and actually had the feeling that it is fruitful in interdisciplinary work to exchange ideas between disciplines, but should not consider ourselves superior to those who start to dig into new terrain (from their perspective) – we tend to leave a similarly bleak trace if we dare to "lean outside of our own territory. I think it is the strength of interdisciplinary researchers that they take the risk to be considered a non-savant outside their area of profound expertise, and we should restrain from spreading bad marks to others from other disciplines (this relates mostly to lines 394–395, 410, 415–424).**

The author is aware that some readers find his writing style to be offensive, has worked hard to try to correct this problem, and is not surprised to find that it persists. There can be little doubt regarding the benefits of exchanging ideas between disciplines, and any specific recommendations would be very welcome regarding how to reword passages so as to be more neutral.

However, having said that, the author is unwilling to refrain from criticizing theories or procedures that are incorrect. The objective here is not to consider anyone as superior, but rather to discover and defend the truth, and the author has taken great care to do this based solidly on fundamental physical laws. From the author's point of view, Jarman (1974) neglected momentum conservation, erroneously classified the description of Parkinson and Penman (1970) as "incorrect" (twice) or "substantially incorrect" (twice more), and misled an entire community of scientists along a mistaken path for several decades. The key question then is how correct this error and prevent its further propagation.

*In an attempt to comply with Dr. Eugster's suggestion and avoid defaming a particular discipline, the author has adopted the following changes*

- *Lines 419: delete ",as has been neglected by the discipline of plant physiology, or ecophysiology";*
- *Line 434: delete ", but broadly neglected in the field of ecophysiology"*
- *Line 447: delete "among plant physiologists"*

**4. The conclusions end with a very general take-home message, but since the author puts so much emphasis in his text to educate plant ecologists, it would be beneficial to have a more specific recommendation set for what plant ecologists finally are supposed to do with this new-gained knowledge. This does not explicitly become clear and the paper would benefit by having such explicit, specific recommendations that I and other could easily pik up, understand, and implement in our own calculations.**

Unfortunately, the author has not been able to develop a "quick fix": an algorithm or equation that would immediately correct WUE or $c_i$ by accounting for non-diffusive transport. This therefore falls under the category of future research, and is open to any scientist who may have better ideas about solving these tricky issues. *The author has made no changes as a result of this comment.*

**"Minor technical issues":**
**L. 86: add "vertical" before velocity**

*The author has made this change (now line 89).*

**L. 403: this appears to be the old notation of the previous (internal) version and should now read (w $|_0$)**

*The author has made this change (now line 427). The author found and corrected the same problem in the last paragraph of the discussion (now line 505).*

**Replies to Anonymous Referee #3**

**General comments: Vertical velocity has a tiny magnitude near surface and is difficult to measure because its magnitude is usually smaller than errors. However, vertical velocity plays a substantial role in mass and energy exchanges between land and atmosphere. For simplicity, they usually assume it is zero at surface. The author argues that it is non-zero by a "thought experiment". The author is a theoretical thinker. This paper shines light on this knowledge gap. I recommend it to be published with minor revision.**

The author thanks the referee for this endorsement. The *author has made no changes as a result of this comment.*

**Specific comments:**

**(1) 2.1.2 The 0th Law of Thermodynamics – I do believe that this is a case from second law of thermodynamics (Postulate of Clausius, see Thermodynamics by Enrico Fermi, 1936). I don't think that "The 0th Law of Thermodynamics" is independent from second law of thermodynamics. So I suggest using the second law of thermodynamics instead of the 0th Law so that your statements no matter heat transfer and mass diffusion are govern by the same second law of thermodynamics. Fourier's law and Fick's law are empirical relationships between fluxes and gradients. Gradients are drivers for fluxes and consequences of fluxes reduce gradients, following a single irreversible direction (entropy increasing) – equilibrium (entropy maximum) –second law of thermodynamics.**

The 1936 textbook cited, although authored by a great physicist, is nonetheless out of date regarding this issue. Modern physics texts (e.g., Giancoli, 1984) recognize that the $1^{st}$ and $2^{nd}$ Laws, although definitively stated first, logically depend on the prior assertion of the $0^{th}$ Law; this explains its odd name. Formalization of the $0^{th}$ Law occurred in the mid-1930s, but was not broadly accepted until well after the publication of Fermi's textbook. The *author has made no changes as a result of this comment.*

**(2) Vertical velocity at surface is always positive (upward) predicted by the equation (4). Based on your thought experiment, this looks true everywhere (leaves, ground, water surface) including large scale (e.g. synoptic scale). To my knowledge, it is sure that vertical velocity is negative in high pressure system areas and positive in low pressure system areas. Therefore, it is difficult for me to understand the positive vertical velocity predicted by your theory in high pressure system areas or divergent air-flow near surface at any scale. Please clarify the conflict in your revision.**

Synoptic-scale velocities are of order 3 cm s$^{-1}$ (e.g., Carlson and Stull, 1986, Subsidence in the nocturnal boundary layer, *J. Clim. Appl. Met.*, **25**, 1088-1099). Whether the boundary condition at the surface is the traditionally conceived $\mathbf{w}|_0 = 0$ cm s$^{-1}$, or $0.000031$ cm s$^{-1}$ as derived from eq. (4), synoptic-scale subsidence implies a convergence in the vertical winds between the surface and the height at which it occurs. It seems that there is no conflict that requires clarification. The *author has made no changes as a result of this comment.*

**(3) Page 6 second paragraph, It is fine to me with "vertical advection" because it is clearly defined by vertical component It does not need to assume horizontal homogeneity.**

The author agrees, and initially proposed simply to delete the last two sentences of this paragraph. However, upon careful reconsideration, the author wishes to finish this paragraph with a statement to clarify a key distinction between advection and diffusion. *Therefore, the author has deleted the two sentences at the end of the first paragraph of section 2.3 (and consequentially the Rannik et al., 2009 paper has been removed from the references section), and added a new sentence to finish this paragraph (now at lines 213-214), to say that:* "
[revised manuscript text omitted]